# Do Molecular Tasks Need Expressive GNNs? Distilling Expressive Models into MPNNs

## Abstract

We investigate the distillation of expressive graph neural networks (GNNs) into simple message passing graph neural networks (MPNNs) for the case of molecular graph data. Under the standard training protocols used in prior work, expressive GNNs substantially outperform simple MPNNs on popular molecular benchmarks. We demonstrate that knowledge distillation closes 50 to 100% of this gap. Importantly, the distilled MPNNs are 2 to 33 times faster than their teachers. Our results suggest that for these molecular tasks, the performance gap is largely due to optimization challenges rather than fundamental expressivity limitations.

## 1 Introduction

It is well-established that message passing graph neural networks (MPNNs) are limited in expressivity: they are unable to distinguish many non-isomorphic graphs (Xu et al., 2019; Morris et al., 2019). Limited expressivity of MPNNs has been given as a motivation for the development of more expressive GNN architectures, which have steadily pushed state-of-the-art performance on molecular graph benchmarks. Fig. 1 shows that expressive GNNs consistently outperform GIN (Xu et al., 2019) – a popular MPNN – on a common molecular benchmark dataset. However, these improvements in predictive performance on molecular tasks often come with computational cost as expressive GNNs frequently have super-linear runtime and rely on expensive preprocessing to compute structural properties of the graph.

Molecular property prediction is not the only area where slow models dominate faster models. In both computer vision and natural language processing, the state of the art is held by large models that only allow for slow inference. For these areas, a common technique to speed up inference is *knowledge distillation* (Hinton et al., 2015; Buciluǎ et al., 2006). In this approach, a slow teacher model with strong predictive performance gets distilled into a faster student model such that the student's predictive performance is similar to that of the teacher. While knowledge distillation has been successful in computer vision and natural language processing, it has not been used for GNNs with

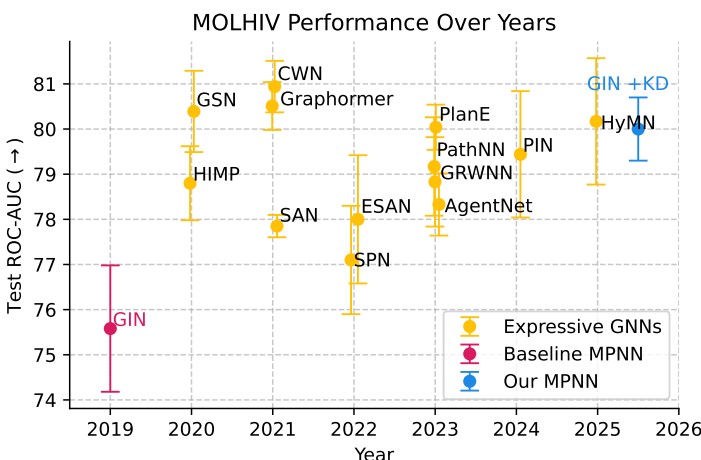

Figure 1: Test performance of different GNNs on the `molhiv` dataset over time. Performance of expressive models was taken from the respective papers (see Appendix A). MPNNs were trained in a standard way (Hu et al., 2020) or with knowledge distillation.

state-of-the-art performance. We perform knowledge distillation from expressive teacher GNNs to less expressive (but faster) student MPNNs. *We demonstrate that this allows us to boost the student's predictive performance to approach that of the teacher while achieving runtime speedups by up to a*

*factor of 33.* Our work also has significant implications on the connection between expressivity and predictive performance of GNNs. In particular, we investigate a fundamental question: to what extent is the performance gap between expressive GNNs and MPNNs on molecular graphs attributable to expressivity limitations versus optimization challenges? Our results with simple knowledge distillation techniques – such as training on teacher logits or teacher-labeled augmented data – suggest that much of the performance gap on molecular graphs stems from optimization difficulties rather than fundamental expressivity limitations, challenging the argument that has been historically used to motivate the proposal of expressive GNNs.

As teachers, we investigate numerous different expressive GNNs: MPNNs with expressive node features (Bouritsas et al., 2022), subgraph GNNs (Bevilacqua et al., 2021), topological message passing (Bodnar et al., 2021), higher-order message passing (Morris et al., 2019), and a graph transformer (Ma et al., 2023). To demonstrate the utility of knowledge distillation, we need to select datasets where expressive GNNs achieve strong predictive performance. We focus on *molecular benchmarks* for two reasons. First, expressive GNNs achieve state-of-the-art performance on molecular datasets (Bodnar et al., 2021; Ma et al., 2023). Second, expressive GNNs struggle to achieve strong predictive performance on non-molecular datasets. Recent work by Coupette et al. (2025) shows that the graph structure is uninformative on many non-molecular benchmark datasets, implying that increased expressivity is not useful (and might even be harmful due to overfitting). Indeed, our own literature research finds that on popular non-molecular datasets, expressive models almost never significantly outperform MPNNs (see Appendix A). Thus, we investigate the predictive performance of these expressive models across a variety of molecular datasets and find that without knowledge distillation *there is a statistically significant gap in performance between expressive models and MPNNs.*

It might seem counter-intuitive to perform knowledge distillation to less expressive models. After all, by definition they cannot learn the same functions as a more expressive model. However, it has been demonstrated that on many popular molecular benchmarks all graphs can be distinguished by MPNNs (Kriege et al., 2020; Zopf, 2022). In principle, this means MPNNs *should* be able to represent functions that match the performance of more expressive GNNs given enough parameters. Yet in practice they fall short, which suggests that they fail not because of limited expressivity, but due to other factors such as optimization difficulties or inductive biases. Through knowledge distillation, we can reduce the impact of these non-expressivity factors, enabling MPNNs to approximate the functions learned by more expressive teachers. We find that this boosts their predictive performance while retaining the runtime efficiency of standard message passing. Fig. 1 illustrates this clearly: while the MPNN GIN (Xu et al., 2019) is significantly outperformed by many expressive GNNs, distillation enables GIN to reach the state of the art.

Our results demonstrate that knowledge-distilled MPNNs often match the performance of their expressive counterparts, offering a practical solution to the trade-off between accuracy and efficiency in GNN design. Recently, the community has been challenged to study the interplay between expressivity, generalization, and optimization more holistically (Morris et al., 2024). Knowledge distillation offers these more holistic insights into the role of expressivity: since distillation does not increase the expressivity of the student, our findings suggest that the performance gap between expressive GNNs and MPNNs is frequently not caused by expressivity. This challenges the often implicit assumption that developing *more expressive* GNNs leads to better predictive performance.

In summary, our contributions are as follows.

- We propose using knowledge distillation from expressive GNNs to MPNNs as a tool to address the trade-off between predictive performance and inference speed.
- We conduct an empirical comparison showing a significant predictive performance gap between expressive GNNs and MPNNs.
- We show that MPNNs trained with knowledge distillation achieve up to $33\times$ faster inference while closing the performance gap to expressive GNNs by over $50\%$.
- We use knowledge distillation to isolate the effect of expressivity from optimization in GNN performance. Our results suggest that on many popular datasets, the predictive performance gap is not caused by expressivity.

## 2 BACKGROUND AND RELATED WORK

We explain basic concepts and give an overview of related research focusing on knowledge distillation and expressivity. We base our notation on the survey by Jegelka (2022). A preliminary version of this paper with a different focus has been presented at a workshop (Anonymous, 2024).

**Graphs.** We define a graph as a triplet $G = (V, E, X)$ where $V$ is a set of vertices, $E \subseteq \{(p, q) \mid p, q \in V, p \neq q\}$ is a set of edges, and $X \in \mathbb{R}^{d \times |V|}$ is a matrix of node features $X = [x_1, \dots, x_{|V|}]$ where $x_v \in \mathbb{R}^d$ is the feature of node $v \in V$. For a node $v \in V$ we denote the set of all *neighbors* by $\mathcal{N}(v) = \{w \mid (w, v) \in E\}$. Real-world graphs such as the molecules we use in this work also often have *edge features*: features $m_e \in \mathbb{R}^r$ of dimension $r \geq 1$ associated with each edge $e \in E$. Since they can be incorporated with minimal changes (see Jegelka (2022)), we omit them from the notation for clarity.

**Graph Neural Networks.** Graph neural networks (GNNs) are machine learning models designed to learn functions on graphs. The most common type of GNN is the *message passing graph neural network* (MPNN) (Gori et al., 2005; Merkwirth & Lengauer, 2005; Gilmer et al., 2017; Hamilton et al., 2017; Kipf & Welling, 2017). An MPNN contains $\ell \geq 1$ layers and computes an embedding $h_v^{(t)} \in \mathbb{R}^{d_\ell}$ of dimension $d_\ell \geq 1$ for each node $v \in V$ in each layer $t \in \{1, \dots, \ell\}$. This is done by initializing the embeddings with the node features and recursively updating the embeddings based on the previous layers. In what follows, we use $\{\{\cdot\}\}$ to denote a multiset, i.e., an unordered collection that can contain the same object multiple times. For each node $v \in V$, we define $h_v^{(0)} = x_v$ and

$$h_v^{(t)} = f_{\mathrm{up}}^{(t)}\big(h_v^{(t-1)}, \{\{h_u^{(t-1)} \mid u \in \mathcal{N}(v)\}\}\big), \ 1 \leq t \leq \ell.$$

Here, the update function $f_{\mathrm{up}}^{(t)}$ is defined as a combination of multilayer perceptrons (MLPs) together with pooling functions (commonly the sum or mean operation). The final output of this MPNN architecture are node embeddings $h_v^{(\ell)}$ for $v \in V$ which can be used for prediction tasks. To obtain a graph representation we can pool the node representations, for example by summing them or computing a mean. The specific MPNN architecture we will use in this work is the *graph isomorphism network* GIN (Xu et al., 2019).

**Expressivity.** Graphs are defined on unordered sets of nodes and defining a canonical ordering is assumed to be computationally hard (Babai, 2016). This means that isomorphic graphs may be stored with different node permutations. GNNs are designed to be permutation invariant which means that they compute the same prediction for all isomorphic graphs. Hence, if a GNN computes different

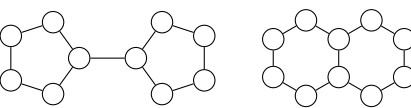

Figure 2: Two graphs that cannot be distinguished by MPNNs.

embeddings for two graphs, these two graphs cannot be isomorphic. The ability of GNNs to distinguish pairs of non-isomorphic graphs is referred to as *expressivity* and is measured by the set of graph pairs that can be distinguished. Xu et al. (2019) and Morris et al. (2019) have shown that the expressivity of MPNNs is limited by the Weisfeiler-Leman graph isomorphism test (WL) (Weisfeiler & Leman, 1968). Note that this limitation persists even if the MPNN has an infinite number of layers, weights, and training data. As a result, MPNNs are unable to distinguish many simple graphs (see Figure 2) and compute many properties that are important for molecular predictions, such as cycle counts (Chen et al., 2020). To improve predictive performance – mainly for molecular tasks – researchers have developed more expressive GNNs and demonstrated that they outperform MPNNs. Appendix A gives an overview of expressive GNNs.

**Knowledge Distillation.** Knowledge distillation was introduced by Buciluă et al. (2006) to compress a large ensemble of models, the *teacher*, into a single *student* model. This is done by training the student on a large amount of artificial data labeled by the teacher. As demonstrated by Hinton et al. (2015), it is possible to distill large teacher neural networks into smaller student neural networks. While knowledge distillation has been applied to GNNs, there are two fundamental differences from our work. First, existing work has focused on reducing the number of parameters in a model (Yang et al., 2022) or avoiding using parts of the data such as edge features (Joshi et al., 2022) or even the entire graph structure (Zhang et al., 2022). Second, existing work often uses complicated distillation methods that are for example based on label propagation Yang et al. (2021), contrastive learning (Joshi et al., 2022; Yu et al., 2022), adaptive temperature (Guo et al., 2023) or neural heat kernels

(Yang et al., 2022). In contrast, we distill expressive GNNs with a high runtime complexity to less expressive linear time MPNNs. Furthermore, while our distillation procedure is simple it still yields strong predictive performance which avoids the complexity and overhead of more intricate methods. To the best of our knowledge, we are the first to perform knowledge distillation between models of different expressivity and use knowledge distillation to analyze the impact of expressivity on the predictive performance of GNNs. For more details on knowledge distillation for GNNs, we refer to the surveys Gou et al. (2021) and Tian et al. (2023).

## 3 BENCHMARKING EXPRESSIVE GNNS AND MPNNS IMPLIES A PERFORMANCE GAP

Knowledge distillation requires strong teacher models. To find the strongest teacher for each dataset, we benchmark different expressive GNNs and an MPNN on molecular datasets. We establish that on most datasets, there exists a statistically significant predictive performance gap between expressive GNNs and MPNNs under a fair evaluation regime. In Section 3.1, we give an overview of the expressive GNNs used in our study. In Section 3.2, we describe our experimental setup. Finally, Section 3.3 describes the results of our benchmark and its implications.

### 3.1 MODELS AND SOURCES OF EXPRESSIVITY

Table 1 lists all GNNs we use in this work. As our baseline MPNN we select the widely used graph isomorphism network architecture (Xu et al., 2019, GIN) due to its expressivity. This MPNN variant is notable, as it can achieve *exactly* WL expressivity, compared to other MPNNs which are less expressive than WL such as GCN (Kipf & Welling, 2017) and GAT (Veličković et al., 2018). As expressive GNNs, we select five different models that extend message passing in distinct ways. GSN (Bouritsas et al., 2022) is an MPNN with expressive node features that encode cycle counts. DSS (Bevilacqua et al., 2021) performs message passing across different subgraphs extracted from the original graphs. CWN (Bodnar et al., 2021) performs message passing on regular cell complex lifts of the input graphs. L2GNN (Morris et al., 2019; 2020b) is a local higher-order WL variant. Finally, GRIT (Ma et al., 2023) is a graph transformer with expressive positional node embeddings based on random walks. For more details, see Appendix B.

Table 1: GNNs used in this work, ✓ marks models that are more expressive than MPNNs.

| Model | Type | Expr. |
|-------|------|-------|
| GIN | MPNN | ✗ |
| GSN | MPNN + Subgraph Counts | ✓ |
| CWN | Topological GNN | ✓ |
| DSS | Subgraph GNN | ✓ |
| L2GNN | Local 2-GNN | ✓ |
| GRIT | Graph Transformer | ✓ |

### 3.2 DATASETS AND EXPERIMENTAL SETUP

Prior work has primarily evaluated expressive GNNs on molecular graphs and a few non-molecular datasets (Paolino et al., 2024; Morris et al., 2019; Bevilacqua et al., 2021; Bodnar et al., 2021). However, as our own literature research and statistical analysis shows, expressive GNNs rarely outperform MPNNs with statistical significance on non-molecular data (see Appendix A). Furthermore, on many non-molecular datasets the graph structure has little relevance to the prediction task (Coupette et al., 2025) making expressive GNNs ineffective on these datasets. We select medium-sized datasets ($\leq$ 130k graphs) with standard holdout splits to avoid the complexity of cross-validation under hyperparameter tuning. All datasets can be found in Table 2. Our experiments cover ZINC (solubility regression, Gómez-Bombarelli et al. (2018); Sterling & Irwin (2015)), QM9 (isotropic polarization $\alpha$ regression, Wu et al. (2018)), alchemy (regression on 12 quantum mechanical properties, Morris et al. (2020a)), and three OGB datasets (Hu et al. (2020)), including the widely used molhiv (HIV inhibition, binary classification).

Table 2: Datasets used in this work.

| Name | #Graphs | Task |
|------|---------|------|
| QM9 | 130,000 | Regression |
| molhiv | 41,127 | Classification |
| alchemy | 12,000 | Regression |
| ZINC | 12,000 | Regression |
| moltox21 | 7,831 | Classification |
| molbace | 1,513 | Classification |
| molesol | 1,128 | Regression |

Table 3: Test performance of the best validation hyperparameter, averaged over 10 runs. **Bold** results are significantly better ($p \leq 0.05$) than MPNNs. An underline identifies the model with the best mean performance for each dataset. GIN and expressive GNNs are discussed in Section 3; GIN+KD shows the best-performing knowledge distillation method on each dataset developed in Section 4.

| Model | QM9 ($\alpha$) (MAE ↓) | alchemy (MAE ↓) | molesol (RMSE ↓) | molbace (ROC-AUC ↑) | moltox21 (ROC-AUC ↑) | molhiv (ROC-AUC ↑) | ZINC (MAE ↓) |
|---|---|---|---|---|---|---|---|
| GIN | 0.235±0.010 | 0.124±0.003 | 1.120±0.060 | 74.8±4.7 | 75.1±0.7 | 75.8±2.1 | 0.176±0.005 |
| GSN | **0.214±0.002** | **0.113±0.001** | **1.063±0.045** | 75.2±2.0 | **76.4±0.9** | 77.1±1.2 | **0.076±0.003** |
| CWN | 0.235±0.031 | 0.122±0.001 | 1.189±0.053 | **78.6±2.4** | 75.2±0.5 | **77.7±1.6** | **0.082±0.003** |
| DSS | **0.207±0.001** | **0.110±0.001** | **0.894±0.034** | **81.3±1.2** | **76.9±1.0** | 75.8±1.8 | **0.101±0.009** |
| GRIT | **0.208±0.003** | **0.105±0.001** | 0.916±0.025 | 81.4±2.0 | 75.1±0.5 | 76.4±0.9 | **0.092±0.004** |
| L2GNN | **0.188±0.002** | **0.107±0.001** | 0.921±0.040 | 68.3±5.0 | **76.7±1.0** | 76.8±1.0 | **0.065±0.001** |
| GIN+KD | > 0.235 | **0.114±0.001** | **1.025±0.005** | **78.6±0.6** | **77.9±0.2** | **80.0±0.7** | **0.077±0.003** [1] |

We use a unified training procedure across models: batch size 128, a cosine learning rate scheduler, and consistent training epochs chosen in accordance with literature. All models share the same hyperparameter grid (embedding dimension, layers, dropout rate, leraning rate, pooling function); `molhiv` and QM9 use smaller embedding sizes for memory reasons. We grid search 48 combinations, select the best on validation, and retrain 10 times to report mean and standard deviation. More details can be found in Appendix E.

### 3.3 Results: A Significant Performance Gap

Our benchmark enables two types of analysis: (1) evaluating models based on their best hyper-parameters selected on the validation set and (2) evaluating model performance across different hyperparameter configurations. We will start with (1). Table 3 shows the test performance of the best hyperparameter combination for each model averaged over 10 training runs. Using Welch's t-test, we find that on every dataset except `molhiv` the majority of expressive models significantly outperform the MPNN GIN ($p \leq 0.05$). This demonstrates a consistent performance gap between more and less expressive GNNs.

**Observation Performance Gap.** *For GNNs with tuned hyperparameters, there exists a significant gap in predictive performance between expressive GNNs and MPNNs on the majority of datasets.*

Moving on to (2), the performance across different hyperparameters, we observe that in the majority of cases, expressive GNNs significantly outperform MPNNs across hyperparameters (see Appendix F). Furthermore, the expressive GNNs in our experiments generally have a smaller standard deviation of performance across different hyperparameters. This means that hyperparameter tuning is less important for more expressive GNNs. However, this phenomenon seems to depend on both dataset and model. For example, L2GNN's mean performance across hyperparameters is actually worse than that of the MPNN for the majority of datasets. Especially on `molesol`, there exist hyperparameter combinations for which L2GNN performs multiple magnitudes worse than the MPNN. In contrast, on all datasets an average hyperparameter combination of DSS outperforms the average MPNN and has a much smaller standard deviation across hyperparameters. This is exemplified by its performance on `molesol`, where the worst performing DSS model still beats the average MPNN.

**Observation Hyperparameter Sensitivity.** *Most expressive GNN architectures in our study require less hyperparameter tuning than MPNNs.*

## 4 Training Strong MPNNs via Knowledge Distillation

Our benchmark in the previous section demonstrated a significant performance gap between expressive GNNs and an MPNN. In this section, we use knowledge distillation to boost the predictive performance of MPNNs while retaining their linear-time runtime. We find that MPNNs trained with knowledge distillation achieve predictive performance competitive with expressive GNNs but are

---

[1]This result is based on training on real-world data from the 250k variant of ZINC labeled with the teacher. On randomly augmented data, the best performing GIN achieves $0.109 \pm 0.001$.

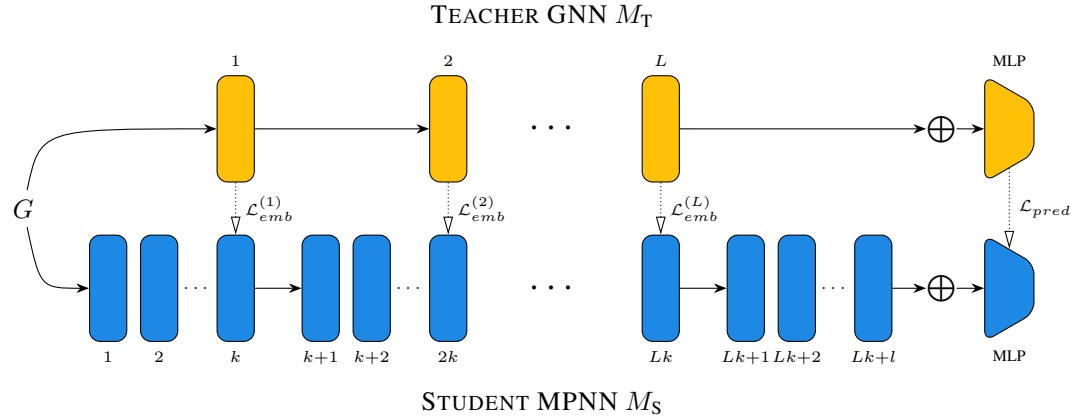

Figure 3: Illustration of knowledge distillation via layer alignment.

significantly faster. We begin by introducing our knowledge distillation procedure in Section 4.1. Then, we separately present our results on classification datasets in Section 4.2 and on regression datasets in Section 4.3. Finally, in Section 4.4 we investigate the impact of knowledge distillation on the predictive performance and inference speed tradeoff.

## 4.1 KNOWLEDGE DISTILLATION

We perform knowledge distillation from an expressive teacher GNN $M_T$ to a student MPNN $M_S$. We assess three different approaches to knowledge distillation: predicting soft rather than hard classes, aligning the embeddings of GNN layers, and augmenting the original training dataset with artificial data. Figure 3 shows our knowledge distillation architecture. Note that knowledge distillation impacts only the training process, leaving the evaluation phase unchanged.

**Soft Classes.** For classification tasks, we can perform knowledge distillation by training a student on the soft distribution predicted by the teacher (Hinton et al., 2015). As all our classification datasets have only binary tasks, we simply train the student via an $L_1$ loss on the teacher's continuous output.

**Layer Alignment.** For regression tasks we cannot use soft class information. Here, we ensure that the student's internal graph representations resemble the teacher's at every layer (see Figure 3). Note that expressive models often compute embeddings for other structures besides vertices. For example, CWN also computes embeddings for edges and cycles. Hence, we compare the mean embeddings at every layer, where the mean is computed over all embedded objects. For an MPNN, this means that we sum the embeddings of all nodes and divide by the number of nodes. We introduce the embedding loss $\mathcal{L}_{emb}$ to align the mean-pooled embedding of the student $M_S$ with the mean-pooled embedding of the teacher $M_T$. For a graph $G$ and GNN $M$, we denote the mean pooled graph embedding produced in layer $i$ as $M^{(i)}(G)$ (details can be found in Appendix E.2). For a teacher $M_T$ with $L \geq 1$ layers we construct a student with $\ell = k \cdot L + l$ layers where $k \geq 1$ and $l \geq 0$ are hyperparameters. We define the embedding loss as the squared Euclidean distance between the pooled embeddings

$$\mathcal{L}_{emb}(M_T, M_S, G) = \sum_{i=1}^{L} \underbrace{\left\| M_T^{(i)}(G) - M_S^{(i \cdot k)}(G) \right\|_2^2}_{\mathcal{L}_{emb}^{(i)} \text{in Fig. 3}}.$$

Finally, we combine the embedding loss $\mathcal{L}_{emb}$ with the *prediction loss* $\mathcal{L}_{pred}$ typical for that task (i.e. the loss used to train the teacher). We use the hyperparameter $\alpha \geq 0$ to balance the importance of these two losses. Let $G$ be a graph, $y$ be the original label on training graphs, or the label predicted by the teacher on augmented graphs (see below). Let $y_S$ be the label predicted by the student. Then, the final loss used for distillation via layer alignment is

$$\mathcal{L}(G, y, y_S) = \mathcal{L}_{pred}(y, y_S) + \alpha \cdot \mathcal{L}_{emb}(M_T, M_S, G).$$

**Training Set Augmentation.** For some regression tasks, layer alignment only slightly improves the student's performance. In these cases, we augment the training set by generating new graphs and labeling them with the teacher. In total, we generate $m$ times as many graphs as in the original training set and say we augment the dataset with a factor of $m$. To generate graphs, we simply modify graphs in the training set by randomly changing features, adding edges, and dropping edges (more information in Appendix E.1).

**Setup.** Unless noted otherwise, we set the hyperparameters of students to that of the teacher for embedding dimension and learning rate. We always use a dropout rate of 0, as knowledge distillation already regularizes the model, and dropout does not work with layer alignment.[2] Furthermore, all students use mean pooling as this is also used by the layer alignment distillation method. As a first step, we train students with the same number of layers as teachers, later we investigate student GNNs with an increased number of layers. For each dataset, we perform knowledge distillation from the best performing expressive GNN model selected on the test set.[3] As knowledge distillation methods differ on classification and regression tasks, we treat these separately in the following sections.

## 4.2    Knowledge Distillation on Classification Datasets

For classification datasets (`moltox21`, `molbace`, `molhiv`), we perform knowledge distillation via predicting soft classes without data augmentation. For each classification dataset, we train the student 10 times and report the mean test set performance. Table 3 compares the results of knowledge distillation against other expressive GNNs and the baseline MPNN GIN. On all datasets, GIN trained via knowledge distillation significantly outperforms ($p \leq 0.05$) the baseline. For `molbace`, knowledge distillation reduces the predictive performance gap by $58\%$ between average performance of the best expressive GNN against the average student. On this dataset, GIN outperforms GSN as well as L2GNN and matches the performance of CWN. For the other two datasets (`moltox21` and `molhiv`), the average student outperforms every other GNN we tested. We believe that part of the reason for the strong student performance is that knowledge distillation acts as a regularizer that reduces overfitting (Hinton et al., 2015): Fig. 4 shows the average test performance as the model trains. For baseline GIN, the performance peaks around 20 epochs at $\approx 78$ ROC-AUC and then converges down to $\approx 75$ ROC-AUC. In contrast, for GIN trained via knowledge distillation, the performance never drops, suggesting less overfitting. Another surprising result is that on average students outperform the teacher. We believe the main reason for this is that we perform knowledge distillation from the single best performing teacher. As a result, students' performance is centered around that single model and not the average across models. Finally, our results are particularly noteworthy, as `molhiv` has been extensively used to test many expressive GNNs. In contrast, our GIN performs *competitively with all expressive GNNs* we could find in the literature (see Fig. 1). Thus, GIN is again state-of-the-art on this dataset.

## 4.3    Knowledge Distillation on Regression Datasets

We apply knowledge distillation to regression tasks. On `ZINC` and `alchemy`, it reduces the performance gap by over $50\%$. For `molesol`, improvements are smaller ($42\%$ gap reduction). On `QM9`, distillation yields no improvements.

**Initial Setting.** Regression tasks lack discrete labels which means that we cannot distill by predicting soft classes. Instead, we apply *layer alignment* with a loss balancing factor $\alpha$. We search over loss balancing factors $\alpha \in \{0.01, 0.1, 1, 10\}$ and data augmentation factors $m \in \{0, 10, 50\}$. We see improvements on all datasets except `QM9`. On `alchemy`, augmentation harms performance, so we exclude both `QM9` and `alchemy` from further scaling experiments. On `ZINC` and `molesol`, augmentation improves results, motivating further scaling experiments.

**Scaling Data.** We increase the augmentation factor $m$ on `ZINC` and `molesol`, using the best loss balancing factor $\alpha$ from grid search. For each $m$, we repeat runs 5 times on `molesol` and 2 times

---

[2]With layer alignment, we compute the difference between the mean student embedding and the mean teacher embedding. As dropout would randomly remove entries from the student embeddings this causes non-convergence and leads to bad student performance.

[3]On some datasets, validation performance is noise and thus not good at predicting test performance (see Appendix F). As we claim that we can distill strong teachers into MPNNs we require teachers with good predictive performance. Hence, we decided to pick teachers based on test set performance.

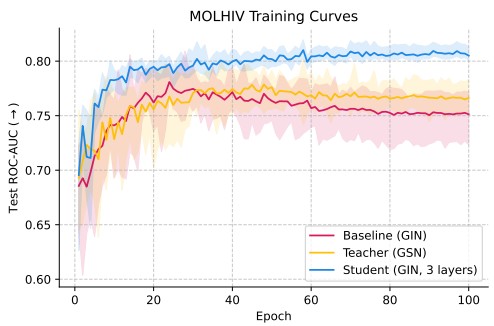

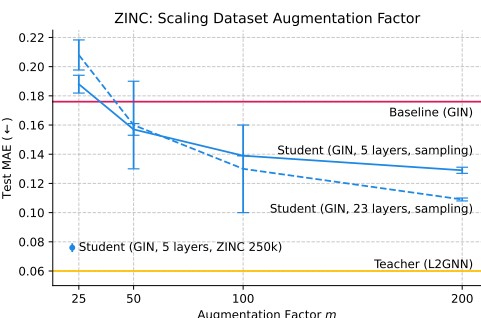

Figure 4: Mean test performance of different models on `molhiv` together with the best and worst performance in each epoch.

Figure 5: Performance of students dependent on the data augmentation for `ZINC`.

on `ZINC`, which is $\approx 50\times$ more expensive to train. On `molesol`, performance peaks at $m = 10$ with RMSE $1.025 \pm 0.005$, closing $42\%$ of the gap. On `ZINC`, performance improves monotonically with $m$ up to 200 (limited by RAM), reaching $0.129 \pm 0.002$ MAE (see Fig 5). This is significantly better than GIN ($0.176 \pm 0.005$, $p \le 0.05$) but still worse than the teacher ($0.065 \pm 0.001$).

**Better Data.** Our previous augmentation method (randomly modifying graphs) is simple and likely suboptimal. To estimate the upper bound of what better augmentation could achieve, we simulate a scenario with almost perfect data augmentation: `ZINC` has a variant containing 250,000 graphs (Gómez-Bombarelli et al., 2018; Sterling & Irwin, 2015). First, we remove all graphs overlapping with the original validation and test sets. Then, we label the remaining data using the teacher model, yielding an effective augmentation factor of $m = 22$. This setup simulates access to abundant, diverse, unlabeled data combined with an oracle teacher. Training on this dataset yields a student MAE of $0.077 \pm 0.003$, nearly closing the gap to the teacher see Fig. 5. This suggests that substantial gains from knowledge distillation are possible if high-quality augmented data is available.

**Reducing Oversmoothing.** Deep GNNs often suffer from *oversmoothing* (Li et al., 2018), where node representations become similar as the network gets deeper. We find that distillation mitigates oversmoothing: On `ZINC`, we train a 23-layer student achieving $0.109 \pm 0.001$ MAE, outperforming shallower students and avoiding the usual MPNN degradation beyond 8 layers (see Fig. 5).

### 4.4 INFERENCE SPEED IMPROVEMENTS

We have seen that MPNNs trained with knowledge distillation can achieve a predictive performance that is competitive with that of expressive GNNs. As expressive GNNs usually have super-linear runtime and MPNNs have linear runtime, this allows us to obtain strong models that are significantly faster. In this section, we benchmark the inference speed of the different models. For each model and dataset, we measure how long it takes to (1) pre-process the entire test set on a single CPU and (2) perform inference on the entire test set with batch size 128. We report a combination of pre-processing and inference speed in Figure 6. For this, we assume that pre-processing is perfectly parallelized over 16 CPU cores. Note that MPNNs do not benefit from this assumption as they do not have a preprocessing step.

For `molhiv`, the distilled three-layer MPNN is between 2 times (GSN) to 14 times (DSS) faster than other expressive models while also achieving stronger predictive performance. For `ZINC`, the distilled five-layer MPNNs GIN + KD trained with generated data augmentation and GIN + KD (250k) trained with "better" data augmentation are faster than all other expressive models. In particular, they are *33 times faster* than L2GNN, the only expressive GNN that significantly outperforms them and are *2.4 times faster* than GSN, the fastest expressive GNN. Figures for the other datasets are shown in Appendix F. Overall, for 6 out of 7 datasets, distilled MPNNs are at least 2.3 times (`alchemy`) to 12 times (`molbace`) faster than expressive GNNs with a higher predictive performance. For details on the runtime overhead of performing knowledge distillation, see Appendix D.

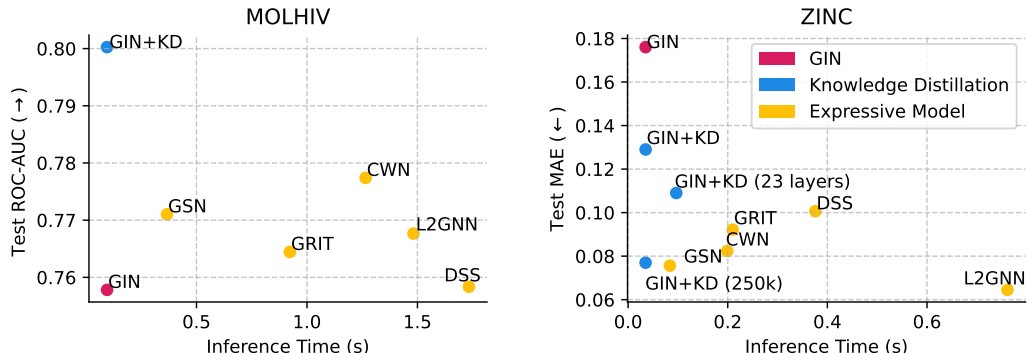

Figure 6: Combined inference speed against test set performance of all models on the test set. Combined inference speed consists of the pre-processing time on a single CPU divided by 16 (to simulate perfect parallelization across 16 cores) plus inference time with batch size 128 on GPU.

# 5 CONCLUSION: SUMMARY, EXPRESSIVITY IMPLICATIONS, AND AN OUTLOOK

**Summary.** We have shown that, on molecular benchmarks, expressive GNNs can be distilled into less expressive MPNNs. This allows us to achieve strong predictive performance while benefiting from the fast runtime of MPNNs. In particular, we have demonstrated that training MPNNs with knowledge distillation improves their predictive performance, often achieving similar performance as expressive GNNs while being 2 to 33 times faster.

**Relation to Expressivity.** Expressivity is an inherent property of the GNN architecture. In particular, MPNNs are always limited in expressivity by WL (Xu et al., 2019; Morris et al., 2019). Thus, even if a teacher GNN is more expressive than WL, a student MPNN is still limited in expressivity by WL. Therefore, in cases where knowledge distillation greatly reduces the predictive performance gap between student and teacher, it follows that this gap *cannot* be due to expressivity. We use knowledge distillation to exclude expressivity from being a cause for the performance gap between expressive GNNs and the MPNN.

**Expressivity Implications.** We can conclude that for many molecular datasets large parts of the performance gap (if not all) are not caused by expressivity. For the widely used ZINC and molhiv datasets distilled MPNNs are competitive with a wide array of expressive GNNs. Our findings challenge the assumption underlying many papers that propose more expressive GNNs to improve predictive performance on molecular data as expressivity is unable to explain the predictive performance of GNNs. New theory is needed to explain this phenomenon (we provide intuition of how distillation and expressivity interact in Appendix C). Furthermore, we have demonstrated that distilled MPNNs can achieve similar predictive performance to expressive GNNs. Without distillation, we have shown that MPNNs fail to match the performance of expressive GNNs. Training strong MPNNs *without* knowledge distillation remains an open challenge. Finally, our results imply that the strong predictive performance of expressive GNNs is based on their inductive bias that are associated with higher expressivity.

**Practical Implications.** Distilling expressive GNNs enables future work in several areas. First, many domains have an abundance of data but only limited amount of *labeled* data. We have shown that in this case, a teacher can be used to label the data and train a faster student with little loss in predictive performance. In domains where data is not abundant, graphs can be generated artificially. In this work, we have only used a simple approach for graph generation which we consider our main limitation as it restricts the quality of training data. More advanced generation techniques may improve distillation performance in low-data settings. Finally, our distillation setup performs better on classification than on regression tasks, indicating that current techniques may not transfer well across task types. Understanding and addressing this discrepancy is crucial for broader applicability of distillation in real-world GNN applications.

**Reproducibility Statement.** We provide details of our experimental setup in Sections 3 and 4. Hyperparameter grids and the data augmentation procedure are described in Appendix E. Our code, together with instructions for reproducing all experiments, is available at https://anonymous.4open.science/r/DistillingExpressiveGNNs.

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

**Use of LLMs in paper writing.** We used LLMs to polish sentences and improve phrasing

## A    CONNECTION TO EXPRESSIVE GNNs IN LITERATURE

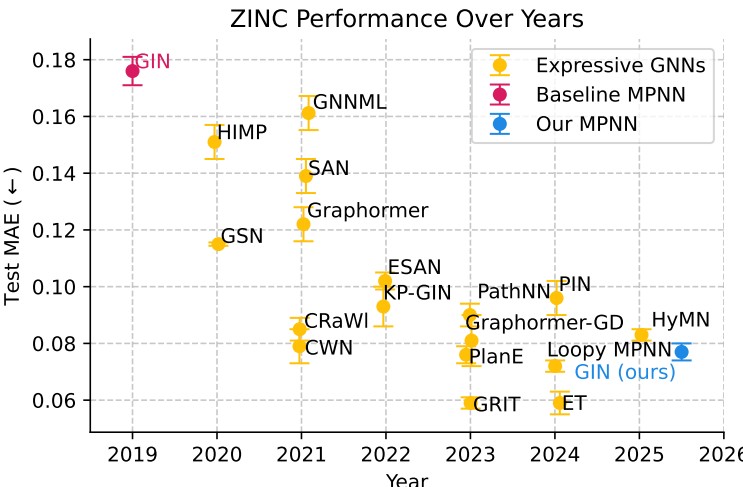

Figure 7: Test performance of different GNNs on the `ZINC` dataset over the year of publication. Performance of expressive models was taken from the respective papers. MPNNs were trained by us in a standard way or with knowledge distillation.

**Predictive Performance.** Figure 1 and Figure 7 show the performance of (expressive) GNNs on `molhiv` and `ZINC` over time. Table 4 lists all GNNs used for this visualization. Note that we do not claim that this is an exhaustive list of all expressive GNNs and that we have changed some of the names to simplify the visualization.

**Non-molecular Data.** To perform knowledge distillation with expressive GNN teachers, we need those teachers to perform well on the chosen datasets. To choose non-molecular datasets, we investigate the datasets used in the works listed in Table 4. In particular, we focus on the three most commonly used non-molecular datasets among these works: `IMDB-M`, `IMDB-B`, and `reddit-B` (Morris et al., 2020a). For all three, we perform a statistical significance test of all expressive models against GIN (Xu et al., 2019). Note that we use the numbers reported in the corresponding works. We find that almost no expressive GNN outperforms GIN on these non-molecular datasets with statistical significance ($p \leq 0.05$). The only exception is KP-GIN (Feng et al., 2022) on `IMDB-B`. Of course, there exist other expressive GNNs proposed in other papers that will probably achieve statistical significance on some of these datasets. However, the broader trend remains: expressive GNNs do not significantly outperform MPNNs on non-molecular data. Thus, we focus on molecular data.

**Related Work.** To improve predictive performance – mainly for molecular tasks – researchers have developed more expressive GNNs and argued that they outperform MPNNs. Such GNNs are often based on pre-computing information that an MPNN cannot detect and injecting this into a GNN. Consider the graphs in Figure 2 that cannot be distinguished by an MPNN. We can pre-compute cycles of length up to 6 and attach to each node a feature that encodes the number of cycles it is part of, making the graphs distinguishable by MPNNs. Common ways of making GNNs more expressive are based on pre-computing subgraph counts (Bouritsas et al., 2022), homomorphisms (Barceló et al., 2021; Jin et al., 2024; Welke et al., 2023) or paths (Michel et al., 2023; Graziani et al., 2024; Paolino et al., 2024). Subgraph GNNs focus on extracting subgraphs and performing message passing across different subgraphs (Bevilacqua et al., 2021; Frasca et al., 2022). There also exist GNNs that are based on $k$-WL and other higher-dimensional variants of WL (Immerman & Lander, 1990) such as $k$-GNNs (Morris et al., 2019) or local $k$-GNNs (Morris et al., 2020b). Finally, transformer based

Table 4: List of GNNs used in Figure 1 and Figure 7.

| Year of Appearance | Method | Reference |
|---|---|---|
| 2019 | GIN | (Xu et al., 2019) |
| 2020 | HIMP | (Fey et al., 2020) |
|  | GSN | (Bouritsas et al., 2022) |
| 2021 | CWN | (Bodnar et al., 2021) |
|  | Graphormer | (Ying et al., 2021) |
|  | SAN | (Kreuzer et al., 2021) |
|  | GNNML | (Balcilar et al., 2021) |
|  | CRaWL | (Tönshoff et al., 2023) |
| 2022 | ESAN | (Bevilacqua et al., 2021) |
|  | KP-GIN | (Feng et al., 2022) |
|  | SPN | (Abboud et al., 2022) |
| 2023 | GRIT | (Ma et al., 2023) |
|  | PlanE | (Dimitrov et al., 2023) |
|  | PathNN | (Michel et al., 2023) |
|  | GRWNN | (Nikolentzos & Vazirgiannis, 2023) |
|  | AgentNet | (Martinkus et al., 2023) |
|  | Graphormer-GD | (Zhang et al., 2024b) |
| 2024 | Loopy MPNN | (Paolino et al., 2024) |
|  | PIN | (Truong & Chin, 2024) |
|  | ET | (Müller et al., 2024) |
| 2025 | HyMN | (Southern et al., 2025) |

GNN architectures (graph transformers) can also be made expressive by selecting a suitable positional encoding (Ma et al., 2023; Müller et al., 2024; Müller & Morris, 2024).

## B  DESCRIPTION OF EXPRESSIVE MODELS

In this section, we describe all expressive GNNs used in our experiments (see Table 1).

**GSN.** The Graph Substructure Network (Bouritsas et al., 2022) extends the MPNN architecture to also include *subgraph counts*. The expressivity of this architecture is at least that of WL but depends on the choice of subgraphs. In our case, we pre-compute the number of non-induced subgraphs forming cycles of length up to 9 and use these as additional features on top of an MPNN. Bouritsas et al. (2022) inject this information on the node and edge level in every message passing layer. We find that only incorporating this information in the initial node features is enough to significantly boost predictive performance[4] and thus use this simplified architecture.

**CWN.** CW Networks (Bodnar et al., 2021) are motivated by the field of *topology* and also pre-compute cycles in the graph. However, edges and cycles in the graphs are added as separate entities instead of adding this information as a feature. This means that CWN computes embeddings for nodes, edges, and cycles via a special form of message passing designed for this topological structure. CWN is strictly more expressive than WL[5] and not less expressive than 3-WL. Our implementation of CWN is based on a graph transformation on top of an MPNN (Jogl et al., 2023) and computes cycles up to length 8.

**DSS.** DSS-GNN (Bevilacqua et al., 2021) belongs to the class of *subgraph GNNs*. It uses a policy to pre-compute different subgraphs and then computes embeddings for all nodes in all subgraphs and passes messages between different subgraphs. The expressivity of this architecture strongly depends

---

[4]For example, our variant of GSN still achieves a 33% lower error on ZINC than Bouritsas et al. (2022).

[5]This means: (1) every pair of graphs distinguishable by WL is also distinguishable by CWN, and (2) there exists a pair of graphs that CWN can distinguish but 3-WL cannot.

on this policy and has been extensively studied (Zhang et al., 2024a; Frasca et al., 2022). Here, we use a policy that computes all 3-hop neighborhoods (3-egonets) selected due to its strong reported performance (Bevilacqua et al., 2021). DSS with this policy is strictly more expressive than WL (Bevilacqua et al., 2021) and upper-bounded in expressivity by 3-WL (Frasca et al., 2022).

**L2GNN.** The Local $\delta$-$k$ Graph Neural Network (Morris et al., 2020b) is based on a *higher-order WL test*. We choose $\delta = 1$ and $k = 2$, which makes L2GNN as expressive as a variant of 2-WL that incorporates local one-hop information. Instead of computing embeddings for nodes, L2GNN computes embeddings for 2-tuples of nodes $(v, w)$ where $v, w \in V(G)$. Adjacency is defined by replacing one vertex of a tuple with a neighboring vertex in the original graph $G$, i.e., the neighbors of a tuple $(v, w)$ are $\{(v, x) \mid x \in \mathcal{N}(w)\}$ and $\{(x, w) \mid x \in \mathcal{N}(v)\}$.

**GRIT.** The Graph Inductive Bias Transformer (Ma et al., 2023) is a *graph transformer* without message passing. Its source of expressivity is a positional encoding called random walk relative encoding. For each pair of nodes $v, w \in V(G)$ and each $i \in \{1, \ldots, k\}$ this positional encoding computes the probability that a random walk of length $i$ starting at $v$ ends at $w$. We use $k = 21$.

## C  GIVING INTUITION: WHY DISTILLATION DOES (NOT) WORK

We give an overview of how different tasks and their required expressivity can interact with knowledge distillation.

### C.1  CASE 1: SOME GRAPHS IN THE DATASET ARE INDISTINGUISHABLE BY MPNNS

#### CASE 1A: GRAPHS ARE INDISTINGUISHABLE AND EXPRESSIVITY IS REQUIRED FOR THE LEARNING TASK

Suppose a dataset similar to the artifical expressivity datasets mentioned by reviewer 5tip. All nodes in all graphs in this dataset have the same node-degree $\Delta$, the same number of nodes, and there exist no features. The task is to count the number of cycles in each graph. In this case, an MPNN would entirely fail: as all nodes have the same degree the MPNN would compute the same node embeddings in each graph. This is evident from simply looking at the update functions of MPNNs:

$$h_v^{(k+1)} = f_{\text{update}}(h_v^{(k)}, \{\{h_u^{(k)} \mid u \in \mathcal{N}(v)\}\}).$$

Since there exist no node and edge features, we would initialize all node features to the same vector $h^{(0)}$. Hence, after one update step:

$$h_v^{(1)} = f_{\text{update}}(h^{(0)}, \{\{h^{(0)} \mid u \in \{1, \ldots, \Delta\}\}\}).$$

Observe that after one update step the embedding of every $v$ is independent from the graph structure (by induction this also holds for all subsequent layers). It follows that in this case, MPNNs are unable to solve the task at hand. Notably, even with KD, MPNNs are still unable to solve this task: KD does not stop the MPNN from predicting the same embedding for every node. KD is merely a different learning task and does not influence the architectural restrictions of MPNNs leading to identical node representations for all nodes.

#### CASE 1B: GRAPHS ARE INDISTINGUISHABLE BUT EXPRESSIVITY IS NOT REQUIRED FOR THE LEARNING TASK

Suppose an arbitrary graph dataset where the task is to predict a function that only depends on tree-homomorphisms. Since WL and MPNNs can count tree homomorphisms they can represent this function even if (some or even all) graphs are indistinguishable, as any two MPNN-indistinguishable graphs must have the same function value. KD may improve empirical performance of MPNNs by providing additional labeled data and/or richer signals to learn from.

## C.2 Case 2: All Graphs in the Dataset are Indistinguishable by MPNNs

### Case 2a: Graphs are distinguishable but expressivity is required for the learning task

Consider the following dataset. Every graph contains two disjoint subgraphs of: - (1) The first sugraph is either two 3-cycles or a 6 cycle. - (2) The second subgraph is a tree that is unique in this dataset.

The task is to predict whether the combined graph contains a 3-cycle or a 6-cycle. Observe that because of (2), all graphs in this dataset are pairwise distinguishable by MPNNs or 1-WL. However, still the MPNN is completely unable to solve this task as it cannot distinguish the two graphs in (1) — this is one of the famous 1-WL counterexamples. Hence, MPNNs can fit the training dataset (even perfectly) but fail to generalize to unseen data. Even when performing KD, MPNNs will still be entirely unable to distinguish the two cases in (1). Hence, even though the graphs are distinguishable MPNNs with KD cannot generalize.

### Case 2b: Graphs are distinguishable and expressivity is not required for the learning task

Real-world graphs very rarely are as "sanitized" as the graphs in Case 2a: while molecules contain cycles there are often tree-like structures attached to nodes in the cycles. Let us now consider the same dataset as in Case 2a, but instead of having a disjoint tree as a subgraph, we will have multiple unique trees that we attach to the vertices in the cycles. While MPNNs in general are unable to count cycles (Chen et al., 2020) in this case the MPNN can learn a function that can detect the cycles by exploiting the fact the the trees make the vertices in the cycle detectable. However, as this task is very hard, the MPNN might never learn this function. KD can make learning this function easier!

## C.3 Implications

On the one hand, we see that KD **cannot** yield an improvement in generalization performance of MPNNs in the examples where beyond-MPNN-expressivity is required for the learning task (Cases 1a and 2a). On the other hand, KD **can** yield an improvement in generalization performance of MPNNs if the learning task at hand is in fact simple enough to be (in principle) learned by MPNNs, as in Case 1b and 2b. That is, in cases where the target function is representable by MPNNs (Case 1b and 2b), KD may improve generalization, while KD is unlikely to increase generalization performance when this is not the case (Case 1a and 2a). Here the MPNN is forced to give identical embeddings to the key substructures. Distillation cannot break this architectural constraint, so the student cannot represent the teacher's function.

This behavior is independent of the question whether the graphs in the dataset can be distinguished by MPNNs, or not. Given that most molecular graphs in the common benchmark datasets can be distinguished by 1-WL (Zopf, 2022) the experiments in our paper (approximately) correspond to case 2. Similar to our Case 2b, real-world molecules have trees attached to cycles that make tasks related to cycles easier. Furthermore, they have node and edges features that can provide additional information about cycles. Given that many molecular properties depend on cycles with attached trees or informative features, KD helps MPNNs learn functions that exploit these structures.

## D The Cost of Knowledge Distillation

We give an extensive overview of the runtime costs of training our models with knowledge distillation and expensive teachers. Overall, knowledge distillation is efficient and only adds a small overhead on top of tuning and training the expressive teachers. The following numbers are per dataset.

- **Hyperparameters Teacher:** The hyperparameters of every teacher are tuned. This means we train 48 teacher models
- **Hyperparameters Student:** For both regression and classification we do not tune any of the MPNN hyperparameters (such as number of layers), instead we copy the hyperparameters of the best teacher. On regression tasks, we tune the loss balancing factor and the data

Table 5: Hyperparameters for the GNN benchmark in Section 3.

| Parameter | All datasets except QM9 and `molhiv` | QM9 and `molhiv` |
|---|---|---|
| Emb. dimension | $\{64, 256\}$ | $\{64, 128\}$ |
| MP layers | $\{2, 3, 5\}$ | $\{2, 3, 5\}$ |
| Dropout | $\{0, 0.5\}$ | $\{0, 0.5\}$ |
| LR | $\{10^{-3}, 10^{-4}\}$ | $\{10^{-3}, 10^{-4}\}$ |
| Pooling | $\{\text{sum}, \text{mean}\}$ | $\{\text{sum}, \text{mean}\}$ |

augmentation factor. In total, we train 9 student models for regression tasks (not counting evaluation and further experimentation). For classification, we only train a single student model as we tune no distillation hyperparameters. Thus, knowledge distillation does not require significantly more hyperparameter tuning.

- **Data labeling:** The cost of data labeling is negligible, as we only need to run a single forward pass of the expressive GNNs and can even disable gradient computation. Labeling all 10'000 graphs in the training set of `ZINC` with L2GNN takes 30 seconds on a single consumer-grade GPU (RTX 3080). We would like to point out that our labeling code is inefficient as it does not make use of batching multiple graphs and thus, labeling could easily be speed up by a factor of 10 - 100.

- **Data augmentation:** Similarly to data labeling this is negligible (3 seconds for 10'000 graphs based on ZINC on a single CPU core). Furthermore, it can be parallelized over multiple CPU cores.

- **Training under teacher supervision:** Since embeddings and logits are pre-computed, the cost of supervision is also negligible. In the case of classification without layer alignment, distillation is as fast as training without distillation. For regression with layer alignment, the only difference is the adapted loss function which means that in practice the difference to training without distillation is small (less than 10% difference). We measured this for 100 epochs of training and evaluation on `ZINC` over 5 runs:

  - With knowledge distillation (layer alignment, no data augmentation): $97.1 \pm 0.5$ seconds
  - Without knowledge distillation: $90 \pm 0.6$ seconds

In summary, we can see that distillation adds almost no overhead when compared to training expressive GNNs. We would like to point out that there is one exception to the efficiency of distillation: the case of training with significantly more data than the teacher (such as $200\times$ `ZINC`). In this case, the training does take significantly longer. However, this is not because distillation is slow, but because of the large amount of data. We plan to include this efficiency analysis in the appendix of our paper.

# E    EXPERIMENTAL DETAILS

Table 5 contains all hyperparameter grids we have used in this paper. Since the number of epochs is important for model performance, we do not use the same number for all datasets. Instead, we select them based on literature (Morris et al., 2019; Bevilacqua et al., 2021) but reduce some slightly to speed up training. On `ZINC`[6] and `alchemy`[7] we train for 500 epochs, on `QM9` for 200 (Morris et al., 2019), and on all OGB datasets for 100 epochs (Bevilacqua et al., 2021). All

---

[6]On `ZINC`, it is common Bodnar et al. (2021); Bevilacqua et al. (2021) to not have a fixed number of epochs and instead reduce the learning rate whenever the model does not improve on the validation set for 20 epochs. If the learning rate dips below a certain threshold, the training stops. As we intend to keep the training setup consistent across datasets, we do not do this. Instead, train for a fixed 500 epochs and set our learning rate with a cosine learning rate scheduler.

[7]This was initially based on Morris et al. (2020b). However, we later realized that they treat the number of epochs as a hyperparameter.

models were trained on servers with one NVIDIA RTX 3080 and 64 of RAM. We trained on up to 5 such servers at a time (one model per server) and used roughly 3000h of GPU compute for all experiments that made it into this paper (excluding preliminary research). Models were implemented with PyTorch (Paszke et al., 2019) and PyTorch Geometric (Fey & Lenssen, 2019). Experiments were tracked with Weights and Biases (Biewald, 2020). Our code can be found at https://anonymous.4open.science/r/DistillingExpressiveGNNs.

### E.1 KNOWLEDGE DISTILLATION VIA DATA AUGMENTATION

We augment the training set via a simple procedure that generates new graphs by randomly modifying graphs in the training set. It then labels the generated graphs with the teacher. Let $G$ be a graph that we want to mutate. First, we iterate through all nodes and all their features and mutate each node feature with a probability of $0.05$. When we mutate a node feature, we randomly sample from the marginal feature distribution on the training set conditioned on the node degree. Next, we randomly drop edges with a probability of $0.05$ per edge which will give us a new edge set $E$. Finally, we randomly add $\lfloor 0.05 \cdot |E| \rfloor$ new edges to the edge set, for each of which we randomly sample an edge feature from the marginal edge feature distribution of the training set.

### E.2 DETAILS ON LAYER ALIGNMENT

We define $\mathrm{M}_{\mathrm{T}}^{(i)}$, the mean pooled embedding of the teacher in layer $i > 0$.

- **GSN** (Bouritsas et al., 2022) only computes node embeddings $h_v^{(i)}$ for every node in the graph. Hence,
$$\mathrm{M}_{\mathrm{T}}^{(i)} = \frac{1}{|V|} \sum_{v \in V} h_v^{(i)}.$$

- **GRIT** (Ma et al., 2023) is a graph transformer that computes node embeddings. Hence, we can define $\mathrm{M}_{\mathrm{T}}$ identically as for GSN.

- **CWN** (Bodnar et al., 2021) computes embeddings for every node, edge and cycle (up to length $k \geq 3$) in the graph. We use $\mathcal{C}_k(G)$ to denote the set of all cycles up to length $k$ in the graph. Then,
$$\mathrm{M}_{\mathrm{T}}^{(i)} = \frac{1}{|V| + |E| + |\mathcal{C}_k|} \left( \sum_{v \in V} h_v^{(i)} + \sum_{e \in E} h_e^{(i)} + \sum_{c \in \mathcal{C}_k} h_c^{(i)} \right).$$

- **DSS** (Bevilacqua et al., 2021) is a subgraph GNN. From a graph $G$ it extracts a set of subgraphs $\mathcal{S}$. For every subgraph $S \in \mathcal{S}$ and each node in that subgraph $v \in V(S)$ it computes an embedding $h_{v,S}^{(i)}$. Then,
$$\mathrm{M}_{\mathrm{T}}^{(i)} = \frac{1}{\sum_{S \in \mathcal{S}} |V(S)|} \sum_{S \in \mathcal{S}} \sum_{v \in S(V)} h_{v,S}^{(i)}.$$

- **L2GNN** (Morris et al., 2020b) computes embeddings $h_{(u,v)}^{(i)}$ for every pair $(u, v)$ of nodes $u \in V, v \in V$. Hence,
$$\mathrm{M}_{\mathrm{T}}^{(i)} = \frac{1}{|V|^2} \sum_{u \in V} \sum_{v \in V} h_{(u,v)}^{(i)}.$$

Table 6: Mean test performance *across all model hyperparameters*. **Bold** results are statistically better than MPNNs with $p \leq 0.05$.

| Model | QM9 ($\alpha$) (MAE ↓) | alchemy (MAE ↓) | molesol (RMSE ↓) | molbace (ROC-AUC ↑) | moltox21 (ROC-AUC ↑) | molhiv (ROC-AUC ↑) | ZINC (MAE ↓) |
|---|---|---|---|---|---|---|---|
| GIN | 1.209±0.878 | 0.236±0.057 | 1.480±0.266 | 74.6±5.5 | 73.1±2.7 | 75.1±2.6 | 0.356±0.122 |
| GSN | **0.746±0.563** | **0.176±0.055** | 1.604±0.437 | **76.9±4.5** | 73.3±2.6 | **76.5±2.7** | **0.164±0.079** |
| CWN | **0.614±0.435** | **0.169±0.049** | 2.655±1.829 | 73.7±4.8 | 73.6±1.6 | **76.2±2.1** | **0.139±0.040** |
| DSS | **0.246±0.033** | **0.131±0.013** | **0.963±0.125** | **79.8±3.4** | **76.5±0.9** | **76.9±1.1** | **0.167±0.072** |
| GRIT | **0.294±0.061** | **0.121±0.013** | **1.116±0.327** | **78.3±3.6** | **74.5±1.4** | 75.6±2.0 | **0.145±0.041** |
| L2GNN | **0.239±0.043** | **0.127±0.013** | 13.412±28.986 | 74.8±5.3 | 73.4±3.4 | 73.3±5.4 | **0.116±0.047** |

# F    MORE EXPERIMENTAL RESULTS

Table 7: Preparation time (seconds) of processing the test set on a single CPU for every dataset and every model. Note that GIN+KD has no pre-processing cost, as it operates directly on graphs.

| Model | alchemy | ogbg-molesol | ogbg-molbace | ogbg-moltox21 | ogbg-molhiv | ZINC |
|---|---|---|---|---|---|---|
| GIN+KD | – | – | – | – | – | – |
| CWN | 1.725±0.046 | 0.348±0.007 | 0.691±0.020 | 8.547±0.051 | 15.980±0.113 | 2.259±0.022 |
| DSS | 1.173±0.002 | 0.205±0.012 | 0.435±0.003 | 3.120±1.455 | 9.170±0.050 | 2.198±0.016 |
| GRIT | 0.392±0.026 | 0.067±0.002 | 0.094±0.004 | 0.983±0.028 | 2.491±0.055 | 0.640±0.026 |
| GSN | 0.543±0.007 | 0.095±0.001 | 0.165±0.001 | 0.846±0.018 | 4.235±0.017 | 0.872±0.020 |
| L2GNN | 0.480±0.021 | 0.067±0.000 | 0.119±0.000 | 0.984±0.435 | 2.855±0.083 | 0.664±0.024 |

Table 8: Time to perform inference (seconds) (after pre-processing) on the test set of every dataset and for every model. GIN+KD always has the same number of layers as the teacher used on that dataset.

| Model | alchemy | ogbg-molesol | ogbg-molbace | ogbg-moltox21 | ogbg-molhiv | ZINC |
|---|---|---|---|---|---|---|
| GIN+KD | 0.026±0.000 | 0.003±0.000 | 0.004±0.000 | 0.036±0.000 | 0.095±0.001 | 0.035±0.000 |
| GIN+KD (23 layer) | - | - | - | - | - | 0.097±0.001 |
| CWN | 0.038±0.000 | 0.006±0.000 | 0.028±0.014 | 0.106±0.013 | 0.267±0.000 | 0.058±0.015 |
| DSS | 0.135±0.001 | 0.026±0.000 | 0.035±0.004 | 0.376±0.021 | 1.160±0.019 | 0.238±0.000 |
| GRIT | 0.145±0.001 | 0.035±0.000 | 0.043±0.000 | 0.184±0.018 | 0.767±0.008 | 0.170±0.001 |
| GSN9 | 0.027±0.000 | 0.003±0.000 | 0.005±0.000 | 0.038±0.000 | 0.102±0.000 | 0.029±0.000 |
| L2GNN | 0.231±0.001 | 0.052±0.000 | 0.099±0.001 | 1.287±0.018 | 1.304±0.025 | 0.720±0.016 |

Table 7 shows the pre-processing time of the test set for each model and dataset measured on a single CPU and averaged over 5 runs. Table 8 shows the inference time on the test set of each dataset and averaged over 5 runs using batch size 128. Figure 8 shows the combined inference speed on all datasets where knowledge distillation improve MPNN performance. Figure 9 shows the impact of scaling the dataset augmentation factor $m$ on molesol. Figure 10 and Figure 11 visualize the model performances from Table 3 and Table 6, respectively. Table 6 shows the mean test performance of each model across all hyperparameters. Finally, we plot validation against test scores of all hyperparameters in Figure 12. This shows, that on some dataset such as molhiv, the validation score is noisy and not a good predictor for test set performance.

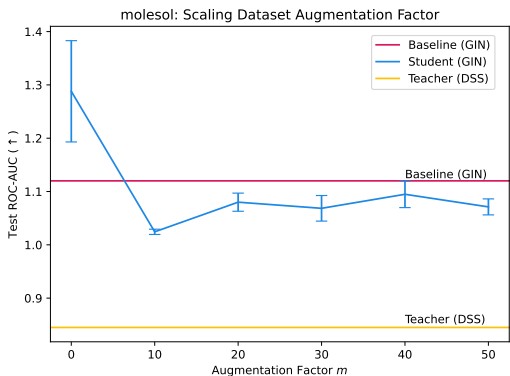

Figure 9: Impact of data augmentation on knowledge distillation performance on molesol.

We report the impact of hyperparameters on model performance in Figures 13, 14, 15, 16, and 17. Some observations:

- Learning rate (Fig. 13): Best learning rate is consistent for each datasets across models.

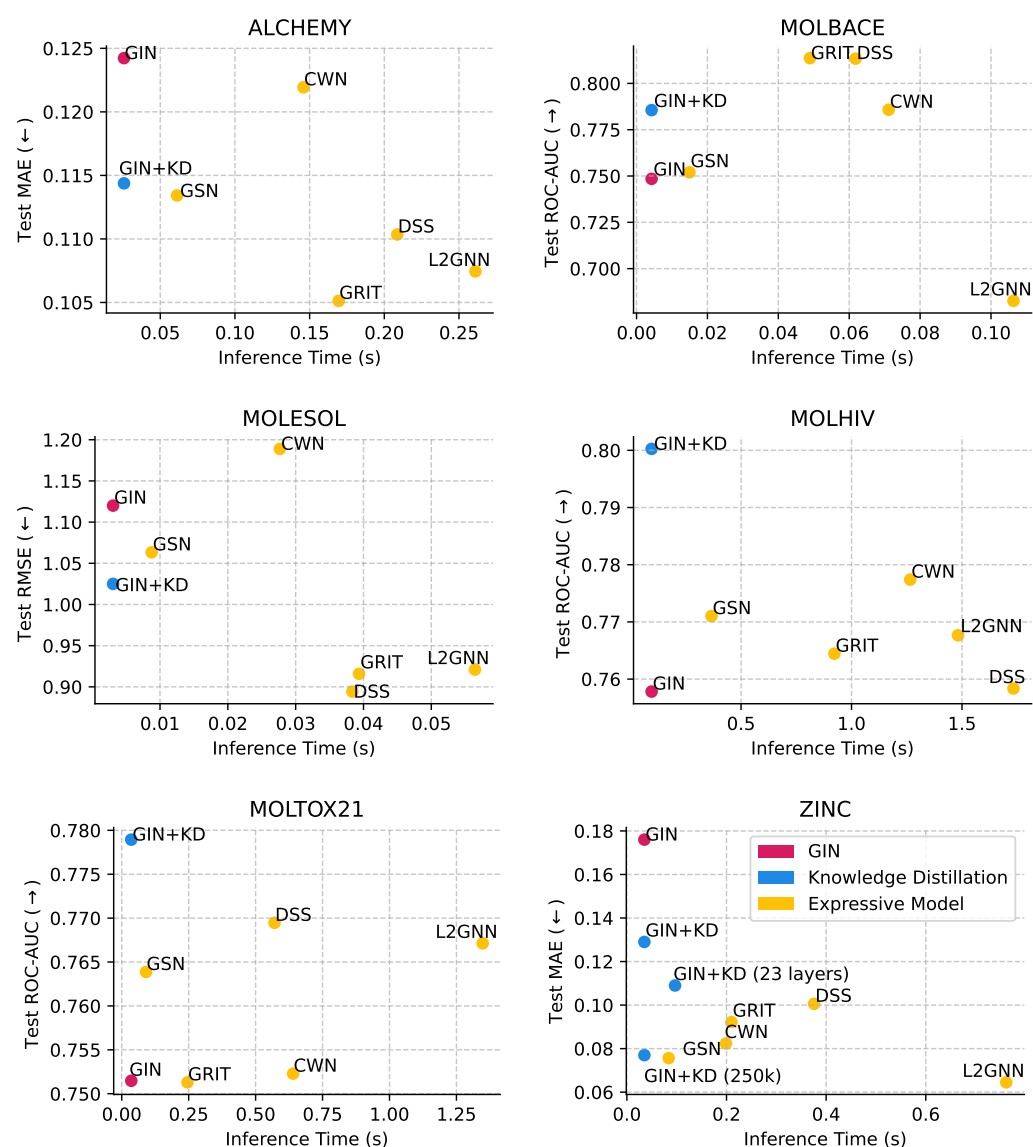

Figure 8: Combined inference speed against test set performance of all models on the test set. Combined inference speed consists of the pre-processing time on a single CPU divided by 16 (to simulate perfect parallelization across 16 cores) plus inference time with batch size 128 on GPU.

- Drop out rate (Fig. 14): For each dataset, the best drop out rate is mostly consistent across models.

- Embedding dimension (Fig. 15): The best embedding dimension is consistent over all models for each datasets.

- Number of layers (Fig. 16): For each `ZINC` and `QM9` more layers are beneficial. For `molesol`, the opposite is true. For all other datasets there are no clear trends.

- Pooling functions (Fig. 17): For many datasets there exists a clear best pooling function that is independent of the GNN.

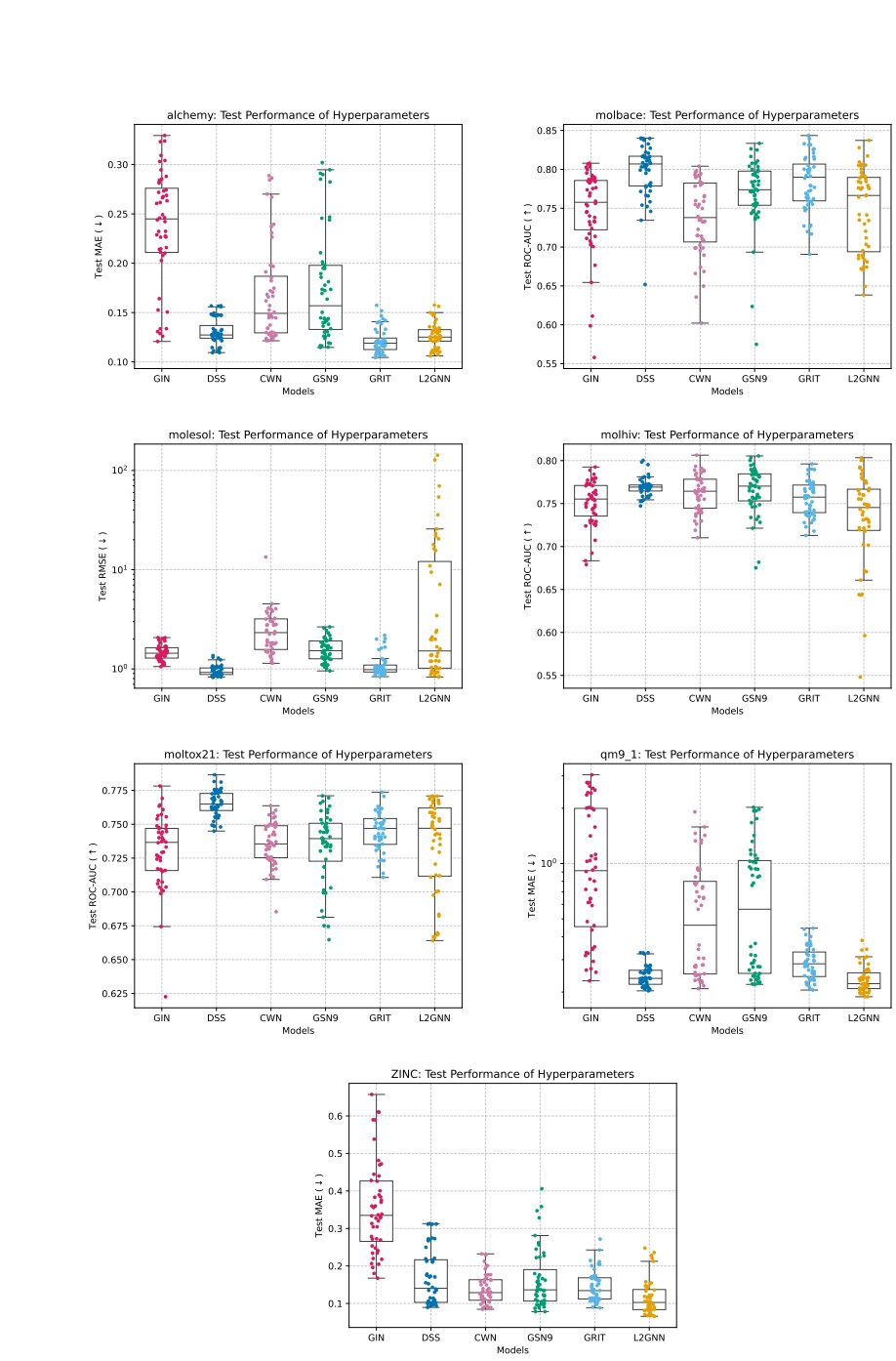

Figure 10: Test set performance of different hyperparameters.

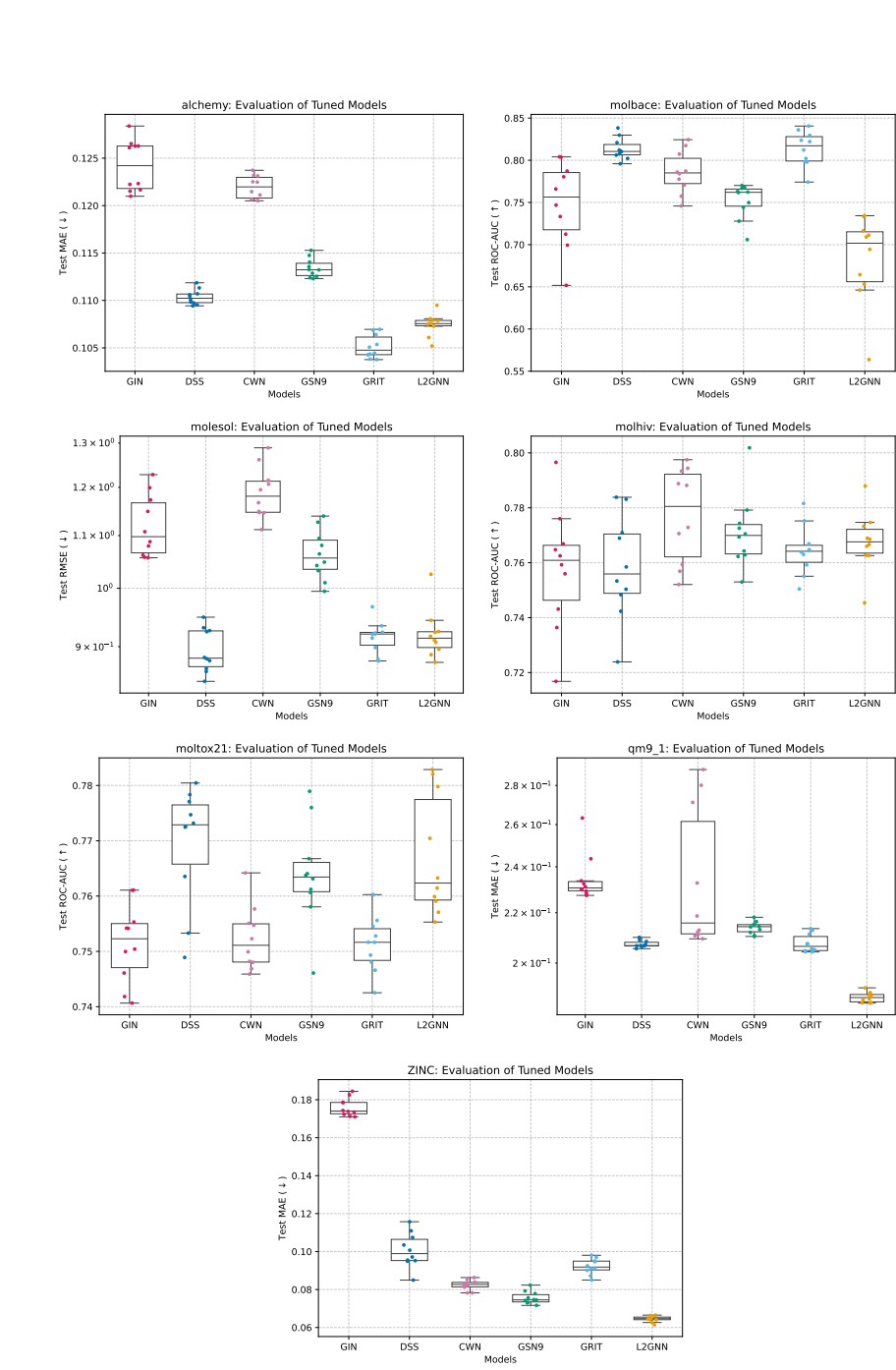

Figure 11: Test set performance of the GNNs with the best hyperparameters.

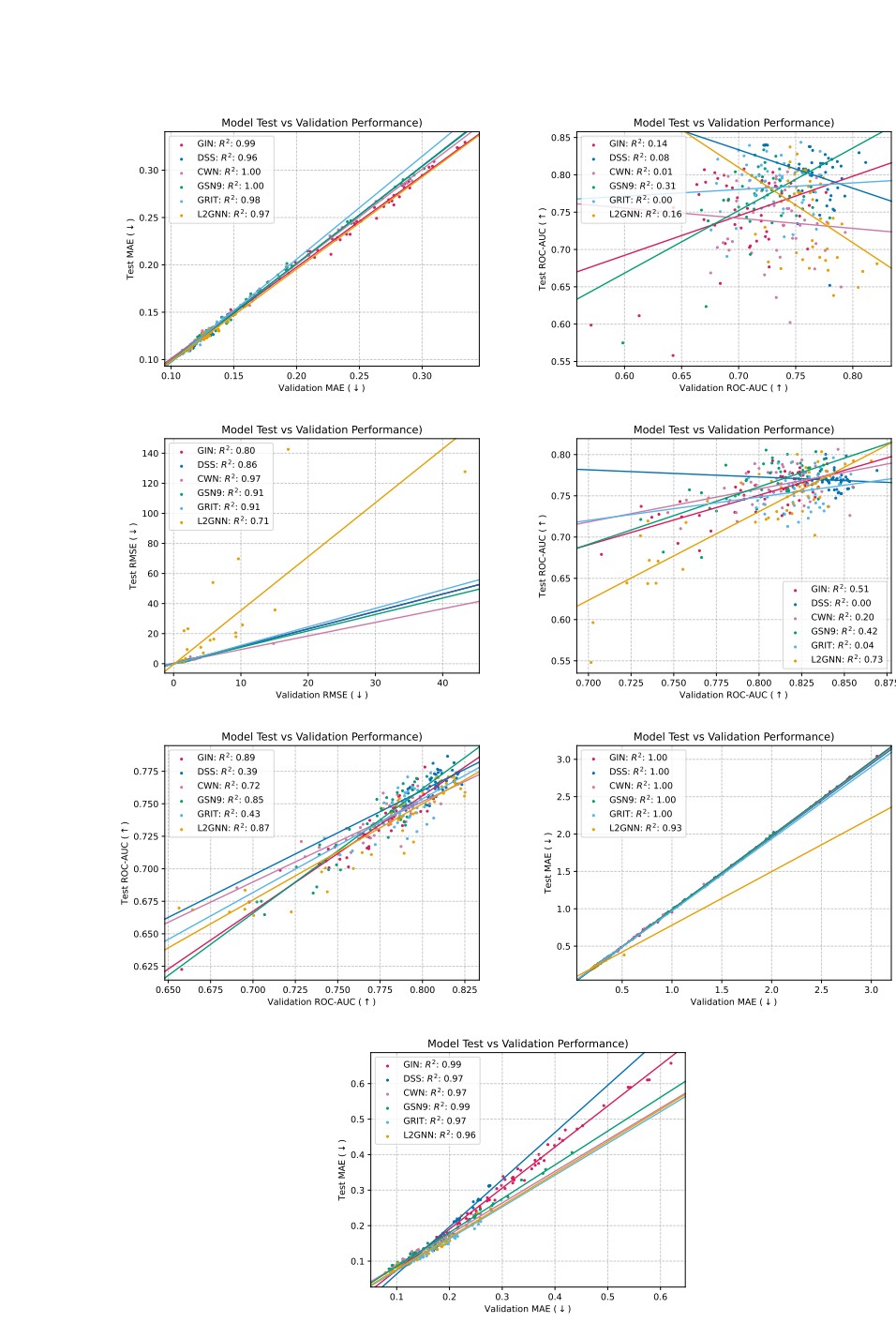

Figure 12: Comparison of test performance against validation performance.

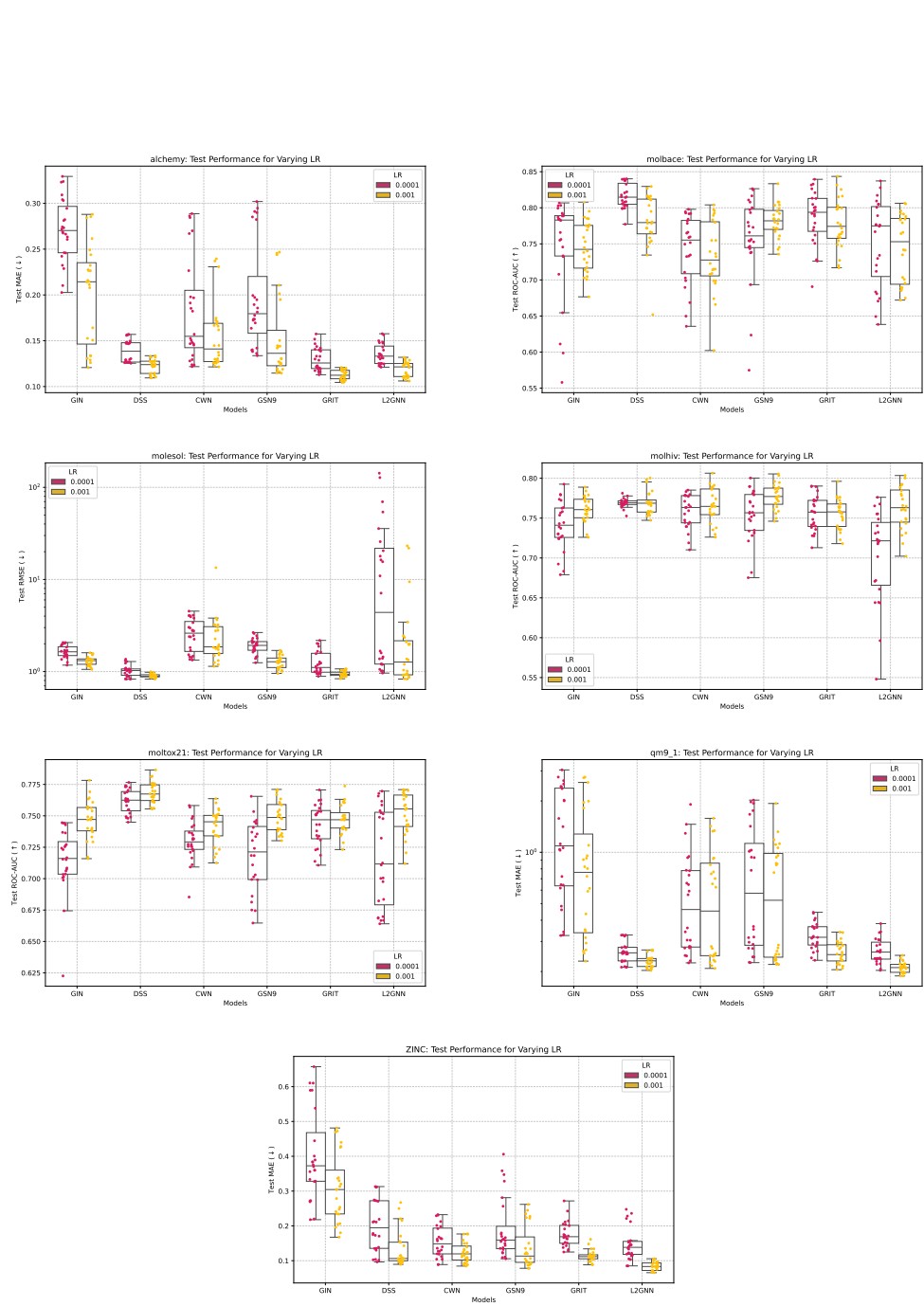

Figure 13: Impact of learning rate (LR) on test performance.

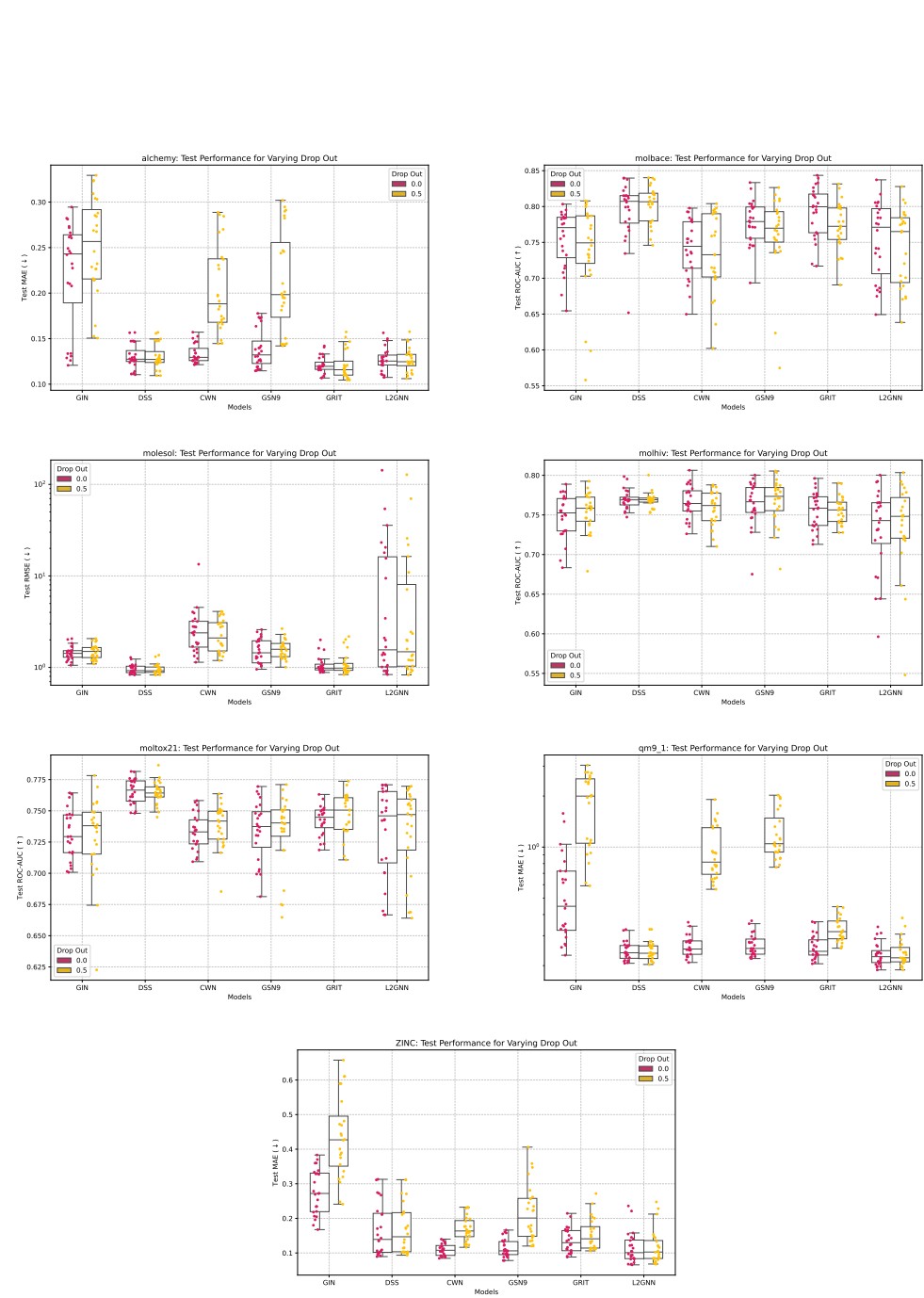

Figure 14: Impact of dropout on test performance.

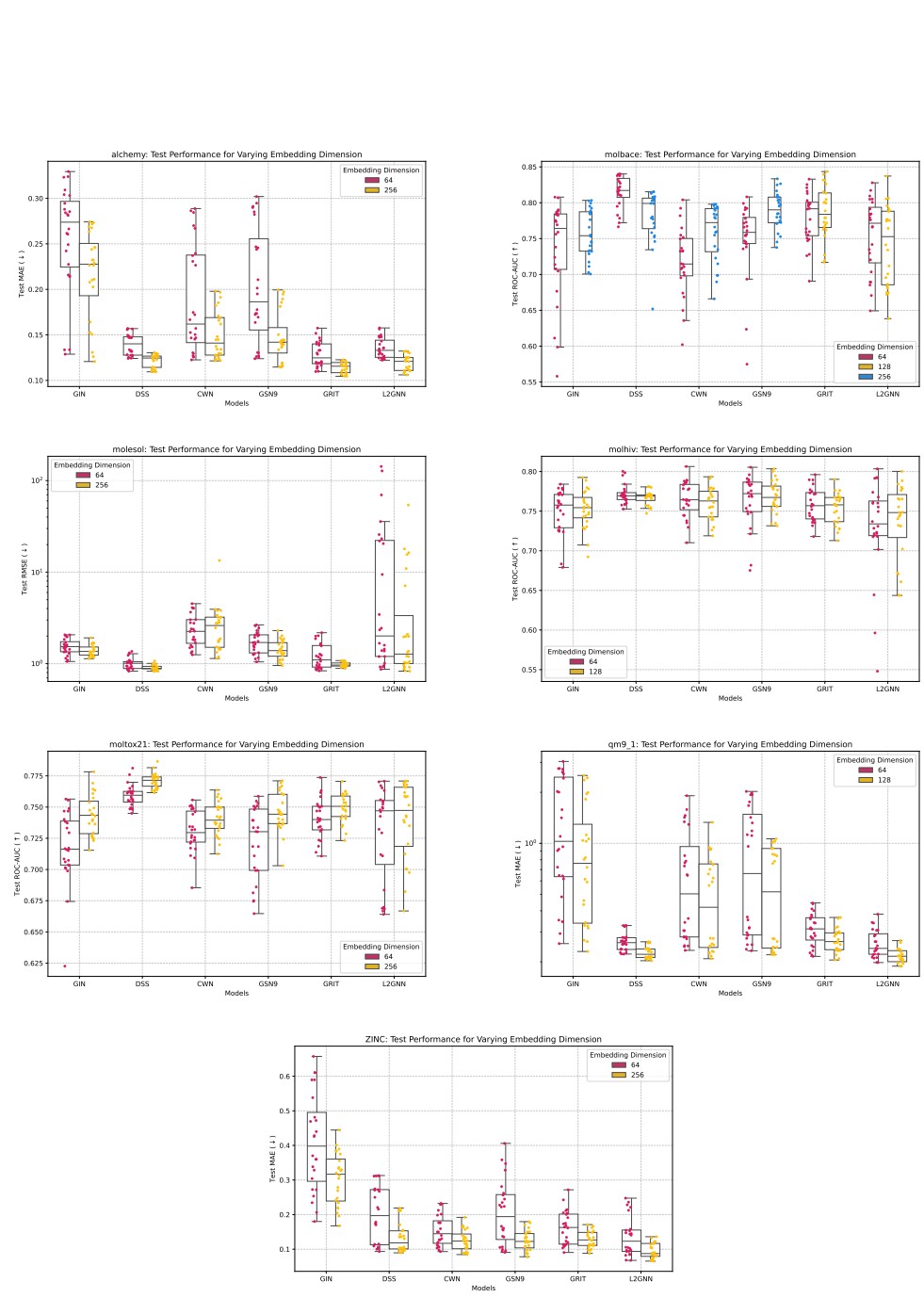

Figure 15: Impact of embedding dimension on test performance.

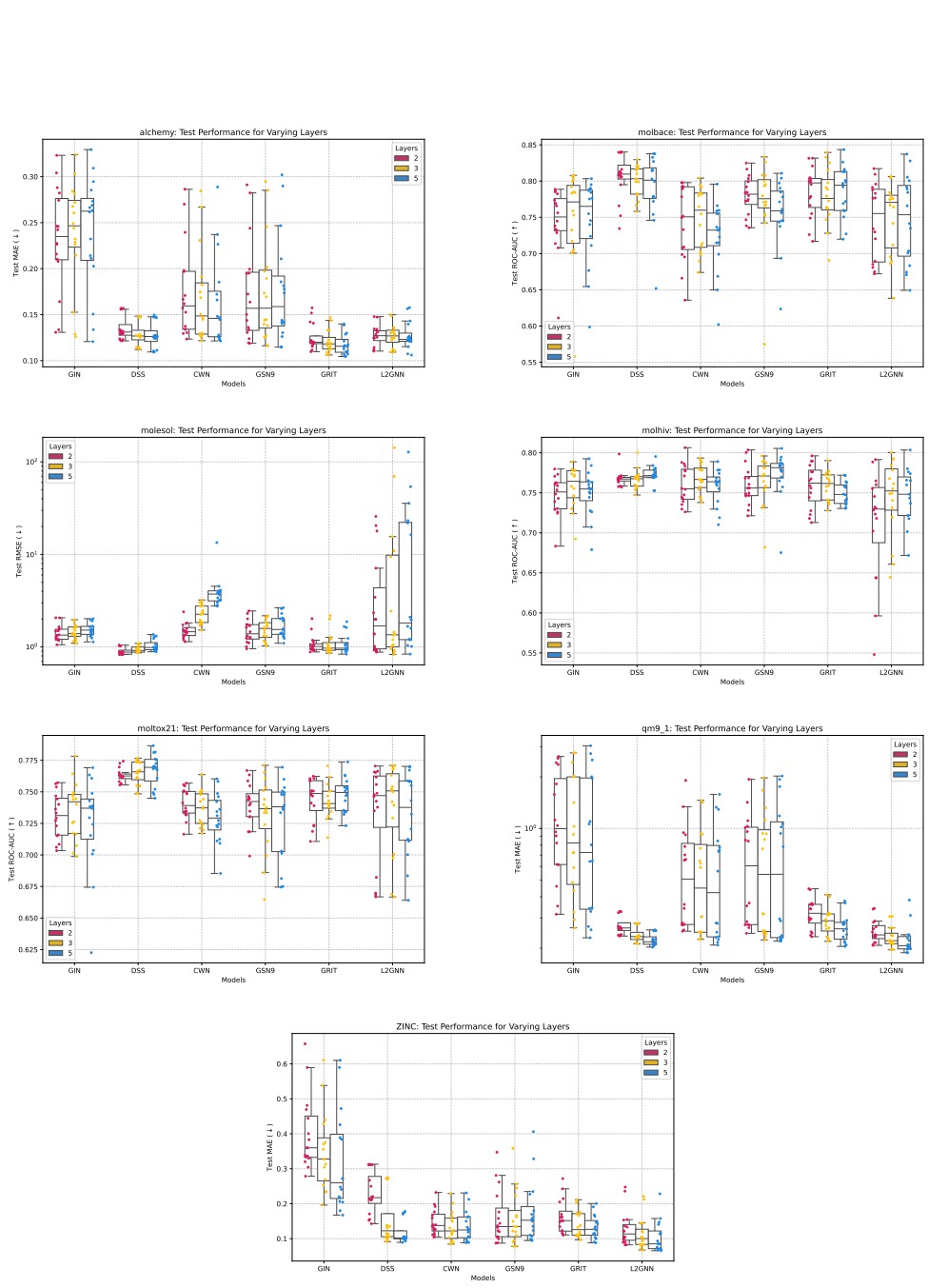

Figure 16: Impact of the number of layers on test performance.

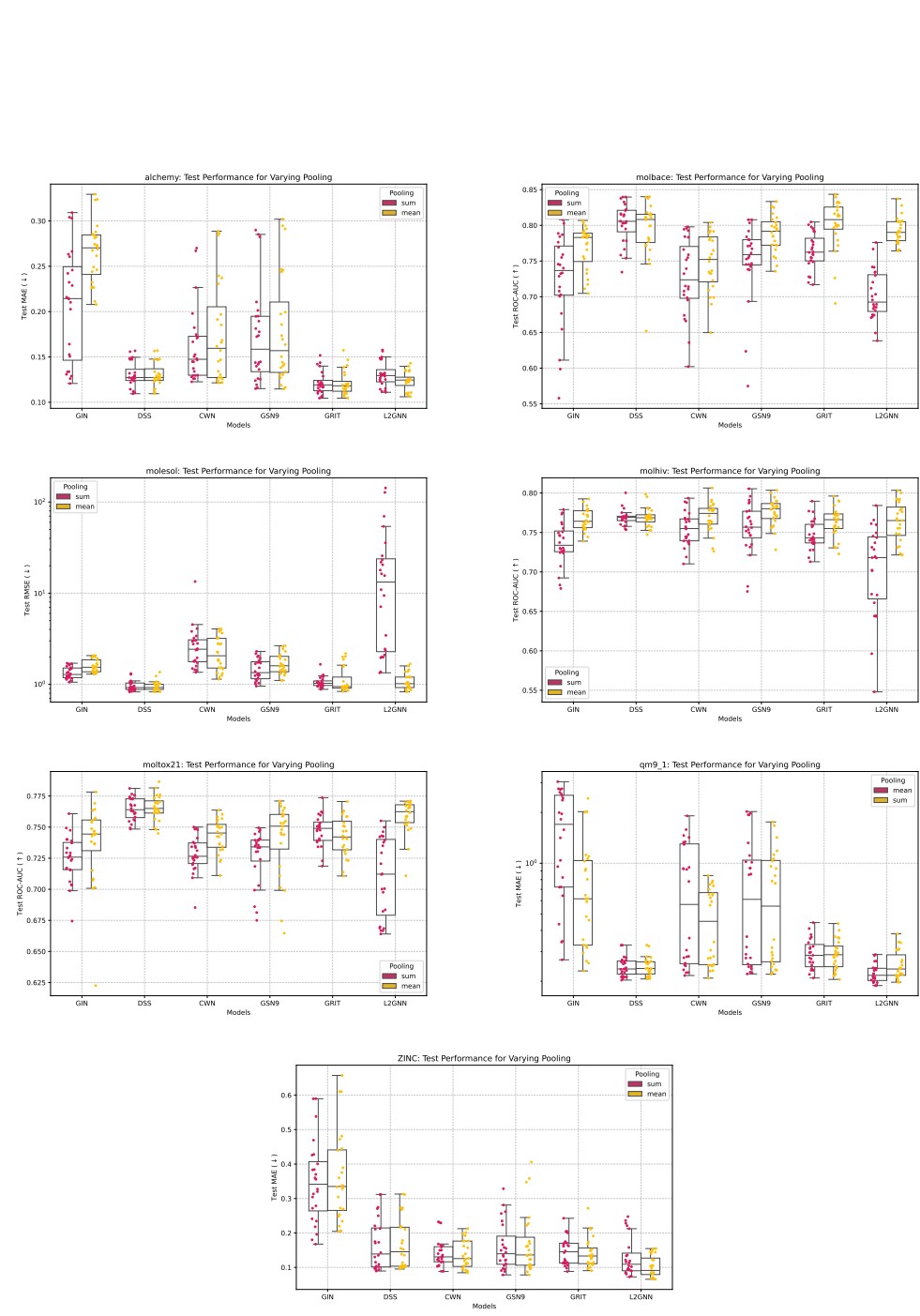

Figure 17: Impact of pooling function on test performance.

