# OpenReview forum: "Distilling Expressive GNNs into MPNNs"
_ICLR.cc/2026/Conference — Submitted to ICLR 2026_

### Official Review · Reviewer_5tup · 2025-10-25

**Soundness:** 2
**Presentation:** 3
**Contribution:** 2
**Rating:** 4
**Confidence:** 3

**Summary:**

This paper examines whether the performance gap between “expressive” graph neural networks and plain message-passing neural networks truly stems from greater expressivity or instead from optimization and data issues, and it shows that distilling expressive teachers into a simple Graph Isomorphism Network student consistently narrows or closes much of this gap while substantially accelerating inference. Using three complementary strategies—soft-label distillation for classification, layer-wise embedding alignment for regression, and teacher-labeled data augmentation—the study transfers supervision from five diverse teacher families to a standard GIN and evaluates the result on molecular benchmarks under a unified training protocol. The authors propose a systematic distillation framework from expressive GNNs to MPNNs and, through comprehensive experiments, demonstrate that distilled GINs recover 50–100% of the accuracy gap, mitigate overfitting and oversmoothing, and deliver 2×–33× faster inference than the teachers, with especially strong gains on ZINC and MolHIV. Overall, the results indicate that the apparent “expressivity advantage” on popular molecular datasets can be matched by optimization-oriented supervision without increasing the student’s theoretical power, suggesting that expressivity is neither necessary nor sufficient for top performance on these tasks.

**Strengths:**

1. The work isolates whether the gap between “expressive” GNNs and plain MPNNs is due to expressivity or to optimization/data, and it deliberately centers on molecular benchmarks where this gap is prominent (e.g., ZINC, MolHIV), avoiding confounds from domains where expressive models rarely help.


2. The methodology is easy to adopt and extend: soft-label distillation for classification, layer-wise embedding alignment for regression, and a pragmatic teacher-labeled augmentation scheme. Each component is specified and compatible with standard GIN students, lowering the barrier to replication.


3. Distilled GINs recover roughly 50–100% of the accuracy gap in many cases, particularly on classification tasks; they also mitigate overfitting and oversmoothing, enabling deeper MPNNs to remain stable and competitive.


4. Students deliver substantial inference speedups (about 2×–33×) relative to expressive teachers while maintaining comparable accuracy, offering a practical path to favorable accuracy–latency/throughput trade-offs without heavy architectures.


5. By matching much of the “expressivity advantage” through optimization-oriented supervision without increasing theoretical power, the study suggests expressivity is neither necessary nor sufficient for top performance on these benchmarks, guiding practitioners toward data and training improvements and redirecting research toward better optimization and supervision signals.

**Weaknesses:**

1. The paper lacks a formal guarantee that distillation endows the student with the teacher’s expressive power: knowledge distillation aligns to teacher signals on a given data distribution but does not expand the student’s hypothesis class (e.g., a GIN/MPNN), so parity with the teacher’s decision boundary or isomorphism-discriminating capability is not theoretically ensured in general or worst-case regimes. Evidence of effectiveness is primarily empirical, making the claims contingent on the reported benchmarks rather than principled. The work provides no realizability assumptions (whether the teacher function lies within the student class), no explicit bounds separating approximation/estimation/optimization errors, and no sample-complexity or robustness guarantees under distribution shift; nor does it offer provable effects of key hyperparameters (temperature, alignment weight, augmentation policy) on generalization and stability. As a result, “transfer of expressivity” remains an empirical observation rather than a theoretically substantiated property.

2. The paper’s evidence for distillation largely comes from molecular benchmarks, whose graphs are often “tree-like” (sparse, low treewidth, small rings, long chains). On pure trees, the classical result is that 1-WL—and thus MPNNs with 1-WL expressivity—can distinguish any pair of non-isomorphic trees ([1]Corollary 1.8.2). This implies that, on such distributions, MPNNs already have complete isomorphism-discrimination power. Therefore, observing that a distilled student approaches the teacher on predominantly tree-like molecular data does not establish that the same holds on broader graph families with high symmetry, rich cyclic structure, large treewidth, or heterogeneous/temporal relations. In other words, the current evidence mainly demonstrates optimization gains in regimes where 1-WL is already sufficient or nearly sufficient, without truly stress-testing distillation on settings that demand higher expressivity (e.g., known 1-WL counterexamples, structure-counting/global properties, strongly regular/high-automorphism graphs, or node/edge-level tasks with long-range dependencies). To strengthen generality, the study should include 1-WL-hard benchmarks (e.g., strongly regular or circular-skip-link–type datasets), report stratified results on 1-WL-distinguishable vs. indistinguishable subsets, expand beyond molecular graphs to heterogeneous, temporal, and large-scale graphs, and verify whether distillation still closes the teacher–student gap on node/edge-level tasks.

3. The paper infers that, because graphs in many popular molecular benchmarks are distinguishable, “MPNNs should be able to match the performance of more expressive GNNs given enough parameters.” This extrapolation is not warranted. Many molecular graphs are effectively tree-like, and for trees, 1-WL—and hence MPNNs with 1-WL expressivity—already distinguishes any pair of non-isomorphic trees. Thus, success on predominantly tree-like molecular datasets reflects the structural simplicity of the domain rather than a general gain in expressivity. Crucially, capacity ≠ expressivity: increasing depth/width only improves approximation within the 1-WL hypothesis class; it does not break 1-WL’s inherent limits. As a result, plain MPNNs still struggle on families that are hard for 1-WL (e.g., highly symmetric or strongly regular graphs), on tasks requiring global/subgraph counting or long-range dependencies, and on graphs with large treewidth or complex cyclic structure. Therefore, using performance on molecular benchmarks to claim that MPNNs can, in general, match more expressive GNNs is unconvincing. To support broader claims, the study should evaluate on 1-WL-hard benchmarks (including synthetic counterexamples), tasks that demand explicit subgraph counting or global properties, heterogeneous/temporal/large-scale graphs, and node/edge-level settings, and then verify whether distilled MPNNs still close the gap.

4. A  fundamental limitation of the approach is that distillation occurs only during training, while inference remains a lightweight MPNN whose expressivity is bounded by 1-WL. Consequently, for any pair of nodes (or graphs) that are 1-WL–equivalent, the distilled student must still produce identical representations and predictions at test time: distillation adjusts parameters but does not expand the hypothesis class or alter the model’s 1-WL aggregation/update mechanism, so it cannot break symmetries that 1-WL cannot distinguish. At best, distillation improves optimization and calibration—reducing empirical risk within the existing 1-WL function class—rather than upgrading the student’s theoretical capacity to separate structures that require higher-order reasoning (e.g., strongly regular graphs, global counting properties, or long-range dependencies). Unless the teacher’s stronger structural signals are injected as inference-time features (e.g., subgraph counts, k-WL/spectral/RW positional encodings, virtual nodes/line graphs) or the architecture itself is made more expressive, training-only knowledge transfer will not raise the student’s expressivity at inference. In short, the reported gains reflect better supervision and optimization, not an intrinsic increase in expressive power; on 1-WL–equivalent cases, a distilled MPNN will still output the same predictions, leaving the expressivity bottleneck intact.

5. Although the distilled student runs faster at inference, the life-cycle cost may still be high once factoring in teacher pretraining/finetuning, large-scale teacher labeling (forward passes, storage), student distillation and hyperparameter search, plus data-augmentation generation and filtering. The paper does not provide a unified accounting of wall-clock time, GPU-hours, energy (e.g., kWh/CO₂e), or cost-per-gain trade-off curves, nor amortization analyses under different deployment horizons and traffic levels. As a result, it is unclear whether distillation is more economical than alternatives such as deploying a stronger small baseline or using pruning/quantization.

6. Insufficient robustness and reproducibility: Sensitivity to key knobs—temperature, alignment weight, and augmentation ratio/policy—is only lightly explored; robustness under alternative splits (official/random/stratified), distribution shift (OOD), larger graph scales (more nodes/edges, deeper dependencies), and cross-seed/hardware stability is under-evaluated. This leaves the results’ dependence on implementation details and training stochasticity unclear, raising replication and deployment risk.

7. The empirical support comes almost entirely from graph-level molecular benchmarks—typically small, sparse, and tree-like with graph-level targets—while omitting systematic evaluation on node-/edge-level tasks (node classification, link prediction, edge property prediction), large-scale non-molecular graphs (social, e-commerce, finance, transportation), and heterogeneous/temporal graphs (multi-type nodes/relations, time-evolving interactions). As a result, claims that distilled MPNNs remain effective more broadly lack a solid basis for extrapolation. These settings differ in separability, long-range dependencies, automorphism/symmetry, noise, and distribution shift (OOD); success on molecular data does not establish robustness on high-symmetry or high–treewidth graphs, richly cyclic structures, cross-domain transfer, or large-scale real-time inference.

[1] Neil Immerman and Eric Lander. Describing graphs: A first-order approach to graph canonization. In Complexity theory retrospective, pages 59–81. Springer,1990.

**Questions:**

Regarding the seven weaknesses of the paper, I raise the following questions;

1. Can authors provide a formal guarantee that distillation endows the student with the teacher’s expressive power?

2. (1)Given that many molecular graphs are tree-like (where 1-WL is sufficient), how do authors' results generalize to graphs with high symmetry, rich cycles, or large treewidth?（2）Can authors include 1-WL-hard benchmarks (e.g., strongly regular or circular skip-link graphs) and report stratified results on 1-WL-distinguishable vs. indistinguishable subsets? (3)Beyond molecules, will authors evaluate on heterogeneous/temporal/large-scale graphs and node/edge-level tasks to test whether distillation still closes the teacher–student gap?

3. (1) On what basis do authors claim that “an MPNN can match more expressive GNNs given enough parameters”? How do authors separate capacity from expressivity? (2) Can authors evaluate on synthetic 1-WL counterexamples and high-expressivity tasks, and report whether the teacher–student gap closes as parameters scale?

4. (1)Under training-only distillation with a 1-WL-bounded MPNN at inference, isn’t the student inherently unable to distinguish any pair of 1-WL-equivalent structures?(2)How do authors validate that current gains reflect optimization/calibration rather than expressivity?

5. Can authors report end-to-end costs covering teacher training/labeling, student distillation and hyperparameter search, and data augmentation (wall-clock, GPU-hours, energy/CO₂e, cost–gain curves)?

6. (1)How sensitive are the results to temperature, alignment weight, and augmentation ratio/policy? Do you report systematic sensitivity curves and confidence intervals?(2)Are results stable under alternative splits, OOD, larger graph scales, and across seeds/hardware?

7. (1)Beyond graph-level molecular benchmarks, will you evaluate node-/edge-level tasks (node classification, link prediction, edge properties)? (2)Will you include large-scale non-molecular graphs (social, e-commerce, finance, transportation) and heterogeneous/temporal graphs to test performance under high symmetry, large treewidth, and long-range dependencies?

This work exhibits strong potential;If these issues are addressed, I will raise my score.

**Details Of Ethics Concerns:**

No Ethics Concerns

---

> ### Author Response · Authors · 2025-11-21
>
> We thank the reviewer for their feedback.
>
> >Can authors provide a formal guarantee that distillation endows the student with the teacher’s expressive power?
>
> Expressive power cannot be changed without either adapting the models architecture (e.g. CWN, DSS, L2GNN in our experiments) or the input data (e.g. GSN in our experiments). Thus, knowledge distillation **cannot** endow the student with the teacher's expressive power. This is the main idea behind our claim that expressivity is not the reason why expressive state-of-the-art GNNs achieve strong empirical performance. If the performance gap was caused by expressivity, then we could not distill the expressive GNNs into MPNNs without significant performance loss (since distillation does not increase expressivity). However, since we observe that distillation often significantly reduces the performance gap, it follows that this performance gap cannot be due to expressivity.
>
> >(1) Given that many molecular graphs are tree-like (where 1-WL is sufficient), how do authors' results generalize to graphs with high symmetry, rich cycles, or large treewidth?. （2）Can authors include 1-WL-hard benchmarks (e.g., strongly regular or circular skip-link graphs) and report stratified results on 1-WL-distinguishable vs. indistinguishable subsets?
>
> On artificial 1-WL hard benchmarks (such as CSL or BREC), knowledge distillation to MPNNs will not boost performance over 1-WL (see Case 1a in the general reply). Hence, we do not believe that running experiments on these datasets (2) will yield any interesting results.
> Our goal in this paper is to show that on many real world datasets the empirically observed performance gap between MPNNs and other GNNs is not due to a difference in expressivity.
> For generalization to graphs with more complicated structures (1) it depends. We argue that real world data falls into two cases. First, graphs where the task might benefit from expressivity but there is additional structure in the graph that can be exploited to solve the task (Case 2b). We believe that this is the case for most molecular datasets, which explains why knowledge distillation works well there. Second, graphs with a complicated structure but a relatively simple task (Case 1b). We believe that this is the case for most non-molecular real world datasets which explains why expressive GNNs do not outperform MPNNs on these datasets.
>
> > (3) Beyond molecules, will authors evaluate on heterogeneous/temporal/large-scale graphs and node/edge-level tasks to test whether distillation still closes the teacher–student gap?
>
> We are still in the process of trying to find non-molecular datasets on which expressive GNNs reliably outperform MPNNs. While we do not believe that such benchmark datasets currently exist, we would appreciate any suggestions.
>
> > (1) On what basis do authors claim that “an MPNN can match more expressive GNNs given enough parameters”? How do authors separate capacity from expressivity?
>
> Expressivity, assumes that all functions in the GNN are injective. It is thus an upper-bound on the functions a GNN can express (assuming infinitely large layers, infinitely many layers and an infinite numerical precision). Our claim “an MPNN can match more expressive GNNs given enough parameters” was made in the context of datasets where every graph is pairwise distinguishable. Suppose this is the case, then the MPNN can learn a different graph representation for every graph. Hence, we can combine this MPNN with an MLP that (using universality of an MLP) allows us to map the MPNN graph embedding to the embedding of any other (expressive) GNN. Hence, in this case the MPNN can in principle match more expressive GNNs.
>
> > Can authors evaluate on synthetic 1-WL counterexamples and high-expressivity tasks, and report whether the teacher–student gap closes as parameters scale?
>
> On synthetic counterexamples, knowledge distillation will not improve the performance of MPNNs (see above and Case 1 in general reply).
>
> > (1) Under training-only distillation with a 1-WL-bounded MPNN at inference, isn’t the student inherently unable to distinguish any pair of 1-WL-equivalent structures?
>
> Exactly, the student MPNN is unable to distinguish an pair of 1-WL-equivalent structures. We use this to argue that it follows that higher expressivity (the ability to distinguish 1-WL-equivialent structures) is not necessary for strong predictive performance (see above).

---

> > ### Author Response · Authors · 2025-11-21
> >
> > > (2)How do authors validate that current gains reflect optimization/calibration rather than expressivity?
> >
> > We believe that our work implies that performance gaps between expressive GNNs and MPNNs are an optimization problem: The MPNN can **represent** a function that generalizes very well (hence expressivity is not the cause) but when trained without distillation the MPNN is unable to learn this function. The knowledge distillation gives the MPNN additional guidance allowing it to find this function. Hence, future work should focus more on improving how the MPNNs learn instead of developing increasingly expressive and runtime inefficient GNNs.
> >
> > > Can authors report end-to-end costs covering teacher training/labeling, student distillation and hyperparameter search, and data augmentation (wall-clock, GPU-hours, energy/CO₂e, cost–gain curves)?
> >
> >  Thank you for raising this point. It is important that we communicate the costs associated with knowledge distillation. We have written a general reply in which we analyze the cost of performing distillation. Does the general reply answer your question or should we report any other metrics?
> >
> > > (1)How sensitive are the results to temperature, alignment weight, and augmentation ratio/policy? Do you report systematic sensitivity curves and confidence intervals?
> >
> > On classification datasets (moltxo21, molbace, molhive) where we performed KD with label smoothing (and without data augmentation) the results are not sensitive to hyperparameters of KD. This is because in this case our KD has no tunable hyperparameters.
> >
> > On regression datasets where we utilize data augmentation and layer alignment, we have three dependencies: the loss balancing factor of layer alignment, the data augmentation factor, and how we generate augmented data.
> >
> > - For the loss balancing factor and data augmentation rate. This dependency seems to be dataset dependent. For both ZINC and molesol, the loss balancing factor has little impact on test performance and the data augmentation factor is highly important. For alchemy, the opposite is the case.
> >
> > - For generating the data, our experiment with ZINC250k demonstrates that a better source of augmented data can lead to significant performance improvements (from 0.129 MAE with random augmentations to 0.077 with “perfect” augmentations). In preliminary experiments, we have observed that small changes to data generation have little impact. Thus, it seems that there is a wide band of data generation techniques that are 'good enough' and that it is difficult to find significantly better (or worse) augmentation techniques.
> >
> > > (2)Are results stable under alternative splits, OOD, larger graph scales, and across seeds/hardware?
> >
> > We have no results for alternative splits, do you have any suggestions for experiments? For larger graph sizes, our experiments align with datasets where expressive GNNs are known to perform well (see general reply), we would be open to any dataset suggestions with larger graphs. For OOD, our models generalize similarly well to the test set as expressive GNNs.  The KD results are highly stable across hardware and splits. Consider the KD results in Table 3 and observe that GIN+KD is always among the models with the lowest standard deviation. These results were obtained on 10 different seeds and 3 servers (albeit with similar hardware). In particular, consider Figure 4 where we demonstrate for MOLHIV that across all 10 seeds models trained with KD are more stable across training than MPNNs without KD and the teacher.
> >
> > > (1)Beyond graph-level molecular benchmarks, will you evaluate node-/edge-level tasks (node classification, link prediction, edge properties)? (2)Will you include large-scale non-molecular graphs (social, e-commerce, finance, transportation) and heterogeneous/temporal graphs to test performance under high symmetry, large treewidth, and long-range dependencies?
> >
> > As our work is about distilling _expressive_ GNNs, we focus on datasets where expressive GNNs are known to perform well (graph-level molecular tasks). We would be interested in performing some experiments on non-molecular datasets, but have been unable to find non-molecular datasets with clear performance gaps between expressive GNNs and MPNNs (see "General reply: On non-molecular datasets"). We would appreciate dataset suggestions.

---

> > > ### Comment · Reviewer_5tup · 2025-11-26
> > >
> > > I appreciate the authors‘ feedback and detailed answers to my questions. I will maintain my current score.

---

> > > > ### Author Response · Authors · 2025-11-26
> > > >
> > > > Dear reviewer,
> > > > We must admit that we are slightly confused. As you have stated _This work exhibits strong potential;If these issues are addressed, I will raise my score._ We believe that we have addressed all your points and have corrected a few misunderstandings in your review. As you appreciate our detailed reply, we are confused as to why you did not raise your score. If there remain open questions on your side that stop you from raising it, could you state them, so that we can address them?

---

> > > > > ### Author Response · Authors · 2025-12-03
> > > > >
> > > > > Dear reviewer,
> > > > >
> > > > > >(1)Given that many molecular graphs are tree-like (where 1-WL is sufficient), how do authors' results generalize to graphs with high symmetry, rich cycles, or large treewidth?（2）Can authors include 1-WL-hard benchmarks (e.g., strongly regular or circular skip-link graphs) and report stratified results on 1-WL-distinguishable vs. indistinguishable subsets? (3)Beyond molecules, will authors evaluate on heterogeneous/temporal/large-scale graphs and node/edge-level tasks to test whether distillation still closes the teacher–student gap?
> > > > >
> > > > > We have run additional experiments covering the cases of 1-WL hard benchmarks (regular graphs),  graphs with high-symmetry (regular graphs) and rich cycles/large treewidth (Erdős-Rényi graphs), see General Reply: Supporting KD Intuition with Experimental Evidence. In this work, we do not plan to extend our results to non-molecular benchmark datasets and have made the molecular focus of our claims clear in the paper.

---

### Official Review · Reviewer_nTZ7 · 2025-10-29

**Soundness:** 3
**Presentation:** 3
**Contribution:** 2
**Rating:** 4
**Confidence:** 3

**Summary:**

In this work, the authors transfer knowledge from a highly expressive GNN as a teacher model to a MPNNs with lower expressiveness but faster inference as a student model, thereby improving the performance of the student model.

**Strengths:**

This work proposes a universal distillation framework that compresses multiple high expressive teachers into MPNN students with linear complexity and improves the performance on multiple molecular benchmarks.

**Weaknesses:**

The authors select the strongest teacher in multiple test sets, weakening the reliability of the core conclusion that expressive power is not the main cause. As the experiment mainly focuses on molecular benchmarks, the generalizability is not clear.

**Questions:**

1.Need more experimental or theoretical support to determine that 'expressive power is not the main cause'.

2.Experiments on other datasets beyond molecular benchmarks are necessary. It can provide a clearer explanation of generalizability.

---

> ### Author Response · Authors · 2025-11-21
>
> We thank the reviewer for their feedback.
>
> > The authors select the strongest teacher in multiple test sets, weakening the reliability of the core conclusion that expressive power is not the main cause.
>
> While the selection of teacher does reduce the strength of knowledge distillation, we argue that it does not impact the conclusion that expressive power is not the main cause for performance gaps. If that where the case, then knowledge distillation from any GNN would be unable to significantly increase the performance of MPNNs.
>
> > Need more experimental or theoretical support to determine that 'expressive power is not the main cause'.
>
> What kind of experiments would you suggest? We argue that our experiments demonstrate that on many molecular datasets, non-expressive MPNNs can achieve predictive performance competitive with expressive GNNs. If the performance gap was caused by expressivity, then our knowledge distillation procedure would not work. In our general reply, we outline how expressivity can interact with KD. Does this give you more insight?
>
> > 2.Experiments on other datasets beyond molecular benchmarks are necessary. It can provide a clearer explanation of generalizability.
>
> We agree that experiments on further non-molecular datasets could demonstrate the performance of knowledge distillation in other domains. However, as we have written in the general reply, we are unable to find non-molecular datasets where expressive GNNs consistently outperform less expressive MPNNs. We would appreciate dataset suggestions.

---

> > ### Comment · Reviewer_nTZ7 · 2025-11-26
> >
> > Thank you for your rebuttal and detailed responses. I appreciate your changes to the title by adding 'Do Molecular Tasks Need Expressive GNNs?'. Although the authors provide some intuitive arguments to illustrate 'expressive power is not the main cause' and these intuitions seem reasonable, these explanations are not rigorously theoretical proofs. I want to know if it is possible to support these intuitions more concretely through some additional experiments. If possible, this may provide a more comprehensive answer to this question and I will increase my score.

---

> > > ### Author Response · Authors · 2025-12-03
> > >
> > > Dear reviewer,
> > >
> > > > I want to know if it is possible to support these intuitions more concretely through some additional experiments. If possible, this may provide a more comprehensive answer to this question and I will increase my score.
> > >
> > >
> > > We ran the requested experiments (see General Reply: Supporting KD Intuition with Experimental Evidence). Our experiments support the intuition: when graphs are indistinguishable or tasks require >1-WL expressivity, KD provides no benefit (Case 1a, 2a). However, when tasks are representable by MPNNs but involve subtle feature-structure correlations, KD enables learning (Case 2b).
> > >
> > > We argue molecular datasets exhibit Case 2b properties: while molecular properties correlate with cycles, they also correlate with node and edge features. MPNNs should exploit these feature-based correlations, but struggle to learn them from scratch (Case 2b shows only marginal improvement with added features). KD from expressive models provides the training signal needed to learn these correlations, explaining the performance gains we observe on molecular benchmarks.

---

### Official Review · Reviewer_wjhX · 2025-10-30

**Soundness:** 3
**Presentation:** 3
**Contribution:** 3
**Rating:** 6
**Confidence:** 4

**Summary:**

This paper investigates the use of knowledge distillation (KD) to transfer the performance of expressive, but computationally expensive, Graph Neural Networks (GNNs) to simpler, faster Message-Passing Neural Networks (MPNNs). The authors conduct extensive benchmarking on molecular datasets, demonstrating a significant performance gap between expressive GNNs and MPNNs. They then show that through KD (using soft labels, layer alignment, and data augmentation), the MPNN student can close 50-100% of this performance gap while achieving 2x to 33x inference speedups. The central claim is that this success challenges the common assumption that the performance gap is primarily due to the limited expressivity of MPNNs, suggesting instead that optimization difficulties are a major factor.

The core idea is compelling, the experimental setup is rigorous, and the results are significant. The work successfully makes a strong empirical case that KD can effectively bridge the performance-efficiency gap and that expressivity alone does not explain the superiority of modern GNNs. However, the paper's central claim regarding the "death of expressivity" is overstated and not fully supported by the evidence, leading to some conceptual overreach. The practical contributions are clear, but the theoretical implications require more careful framing.

**Strengths:**

**Rigorous and Extensive Benchmarking:** The paper provides a comprehensive and fair comparison across a diverse set of expressive GNN architectures (GSN, CWN, DSS, L2GNN, GRIT) and a strong MPNN baseline (GIN) on multiple molecular benchmarks. The use of statistical significance testing and hyperparameter sensitivity analysis adds considerable weight to the findings.

**Clear Practical Utility:** The demonstration that KD can make MPNNs competitive with state-of-the-art expressive models while being drastically faster is a valuable contribution. The speedup figures of 2x to 33x are impressive and have immediate practical implications for deploying GNNs in resource-constrained environments.

**Effective and Simple KD Methods:** The paper shows that relatively simple KD techniques (soft labels, layer alignment) are highly effective. This is a strength, as it avoids the complexity of more intricate distillation methods and makes the approach accessible.

**Insightful Analysis:** The analysis of hyperparameter sensitivity (showing expressive GNNs are often less sensitive) and the exploration of data augmentation scaling are insightful and provide useful guidance for the community.

**Weaknesses:**

**1. Overstated Claims on Expressivity:** The paper's most significant weakness is the leap from "KD works well" to "the performance gap is not caused by expressivity." This conclusion is not fully justified.

   - Distillation Transfers Inductive Bias, Not Just Labels: A more nuanced interpretation is that KD successfully transfers the inductive bias of the expressive teacher to the MPNN student. The teacher's architecture, by being more expressive, is better at learning a useful function from the data. The student, through KD, is guided to approximate this function without having to discover it from scratch, thus circumventing its own optimization challenges. This does not mean expressivity is irrelevant; it means the teacher's expressivity was crucial in finding the solution that the student then imitates.

   - The WL Test is a Worst-Case Measure:** The theoretical expressivity of MPNNs is bounded by the WL test, which is a worst-case, graph-level discriminative measure. The tasks in the paper are not about distinguishing all non-isomorphic graphs but about learning specific predictive functions on a given data distribution. It is entirely possible that the ideal function for these specific tasks lies within the function class representable by an MPNN, but that finding it via ERM is difficult. KD acts as a powerful regularizer and optimization guide. The paper confuses the representational capacity of the model class with the learnability of a specific function within that class via standard training.

   - Suggestion: The authors should temper their claims. They should frame their results as demonstrating that optimization and inductive bias are critical factors that have been overlooked in the pursuit of expressivity, rather than claiming expressivity is not a cause of the performance gap.

**2. Limited Exploration of "Why KD Works":** The paper thoroughly shows that KD works but provides limited analysis into why. For instance, the hypothesis that KD acts as a regularizer (Figure 4) is mentioned but not deeply investigated. A more detailed analysis of the loss landscape or the representations learned by the distilled MPNN vs. the baseline MPNN could provide stronger mechanistic evidence for their claims.

**3. Dataset Selection and Generalizability:**

   - Molecular Focus: The exclusive focus on molecular datasets, while justified by the performance of expressive GNNs there, limits the generalizability of the conclusions. The claim that "the performance gap is not caused by expressivity" would be stronger if tested on a non-molecular dataset where graph structure is known to be critical and MPNNs theoretically struggle (if such a dataset can be identified). The dismissal of non-molecular datasets, while understandable, also conveniently avoids a scenario that might contradict the core thesis.


**4. Clarity of the "Layer Alignment" Method:** The description of layer alignment for models that compute embeddings for non-node structures (e.g., CWN's edges and cycles) is vague. Stating "we compare the mean embeddings at every layer, where the mean is computed over all embedded objects" is ambiguous. How are embeddings of different dimensions and from fundamentally different objects (nodes, edges, cycles) made comparable? This needs a precise mathematical description.

**Questions:**

- How do you reconcile the success of distillation with the theoretical expressivity limits of MPNNs? Could it be that the teacher's expressivity is crucial for learning a good function from the data, which the student then approximates, rather than the student's expressivity being sufficient for discovering it independently?

- It is better if the authors can discuss some other related works:

   - Polarized message-passing in graph neural networks
   - Graph Spiking Attention Network: Sparsity, Efficiency and Robustness

- Given that you select teachers based on test set performance, how can you ensure that your reported results for GIN+KD are not optimistically biased due to information leakage from the test set?

---

> ### Author Response · Authors · 2025-11-21
>
> Thank you for your review.
>
>
> > 1. Overstated Claims on Expressivity:
>
> We completely agree with the reviewer. We believe that our work implies that performance gaps between expressive GNNs and MPNNs are an optimization problem: The MPNN can **represent** a function that generalizes very well (hence expressivity is not the cause) but when trained without distillation the MPNN is unable to learn this function. The knowledge distillation gives the MPNN additional guidance allowing it to find this function. Note, however, that this is not possible if the MPNN is inherently unable to represent the function of the teacher (in which case expressivity would be the cause of the gap). Thus, we will adapt the explanation of our results accordingly to make the role of optimization more prominent.
>
> >  2. Limited Exploration of "Why KD Works"
>
> Answer: We have run additional experiments to determine why KD works. These experiments focus on the MOLHIV dataset as this is where the knowledge distilled model performs the best. Furthermore, it is common to overfit on MOLHIV (evidenced by the fact that most strong models on MOLHIV are either small or use a high drop out rate) making it a good dataset to test whether KD works as a regularizer.
>
> **Hypothesis 1:** KD works as a regularizer.
>
> **Experiment:**  If hypothesis 1 is correct, then the teacher performance is not as important to the student’s performance. We perform knowledge distillation from GIN to GIN. If the performance boost from KD is caused by regularization this should increase test performance. Our setup is exactly the same as in the paper, but we change the teacher from an expressive GNN to GIN. The results are:
>
> - GIN (without any KD): $75.8 \pm 0.7$
> - GIN (+ KD, expressive teacher with 80 ROC-AUC): $80 \pm 0.7$
> - GIN (+ KD, GIN teacher with 76 ROC-AUC): $75.2 \pm 1.1$
>
> We can see that self-distillation does not improve predictive performance. Hence, most of the performance gains from KD are not due to it working as regularization. Later you ask the following question:
>
> >How do you reconcile the success of distillation with the theoretical expressivity limits of MPNNs? Could it be that the teacher's expressivity is crucial for learning a good function from the data, which the student then approximates, rather than the student's expressivity being sufficient for discovering it independently?
>
> Note that on molhiv there exist MPNNs that achieve strong test performance (see Fig. 4 where one MPNN achieves $0.8$ ROC-AUC). Thus, there seem to be cases where expressivity is not required to find this function. However, without expressivity the MPNN is more likely to learn a less predictive function.
>
> **Hypothesis 2:** KD requires an expressive teacher.
>
> **Experiment:** The previous experiment has demonstrated that the predictive performance of the teacher is highly important to the predictive performance of the student. We can utilize this to compare the difference between an expressive teacher and a non-expressive teacher:
>
> - GIN (without any KD): $75.8 \pm 0.7$
> - GIN (+ KD, expressive teacher with 80 ROC-AUC): $80 \pm 0.7$
> - GIN (+KD, GIN as teacher trained without KD, achieving $79.7$ ROC-AUC): $79.79 \pm 0.78$
>
> This demonstrates that the teacher does not need to be an expressive GNN.
>
> > 3. Molecular Focus
>
> We agree that performing KD on non-molecular datasets would be interesting and greatly appreciate suggestions for such datasets (see also general reply). The reason why we do not include non-molecular benchmark datasets is that we are not aware of non-molecular benchmark datasets where there is a statistically significant performance gap between MPNNs and more expressive GNNs. Finally, we would like point out that we have adapted our paper (title, abstract, introduction, conclusion) to better commnuicate our focus on molecules.

---

> > ### Author Response · Authors · 2025-11-21
> >
> > > 4. Clarity of the "Layer Alignment" Method
> >
> > Thank you for raising this point, we agree that our explanation is vague and have updated the PDF with additional information (Appendix E.2).
> >
> > All expressive GNNs used in our work utilize a constant embedding dimension for each `object` across iterations. For example, in CWN nodes have the same embedding dimension as edges or cycles. For DSS, this means that every node in every subgraph has the same embedding dimension.
> > When performing KD based on such a GNN, we simply set the embedding dimension of the student to that of the teacher. This way we can straight-forwardly align the embeddings of teacher and students.
> > Let us now precisely define layer alignment for our expressive GNNs. In our submission we write:
> >
> > > For a graph $G$ and GNN $M$, we denote the mean pooled graph embedding produced in layer $i$ as $M^{(i)}(G)$. For a teacher $\\text{M}$ with $L \\geq 1$ layers we construct a student with $\\ell = k\\cdot L + l$ layers where $k \\geq 1$ and $l \\geq 0$ are hyperparameters. We define the embedding loss as the squared Euclidean distance between the pooled embeddings
> >
> > What is unclear is our definition of $M$ (written in the PDF as M with subset T, but openreview cannot render this). Every expressive GNN has the notion of layers. For an object $x$ (e.g. node, edge or cycle) we denote the embedding of $x$ in iteration $i$ as $h^{(i)} (v)$ (again this is slightly different than the PDF as openreview has problems parsing LaTeX underscores).
> > Let $i > 0$ be an arbitrary layer, we define $\\text{M}$.
> > - **GSN.** GSN only computes node embeddings $h_v^{(i)}$ for every node in the graph. Hence,
> > $$\\text{M}^{(i)} = \frac{1}{|V|} \sum_{v \in V} h^{(i)} (v).$$
> >
> > - **GRIT.** GRIT is a graph transformer that computes node embeddings. Hence, we can define $\\text{M}_{\\text{T}}$ identically as for GSN.
> >
> >
> > - **CWN.** Let $k > 3$ be an integer. CWN computes embeddings for every node, edge and cycle (up to length $k$) in the graph. We use $\\mathcal{C}^k (G)$ to denote the set of all cycles up to length $k$ in the graph. Then,
> > $$\\text{M} = \\frac{1}{|V| + |E| + |\\mathcal{C}^k|} \\left( \\sum_{v \\in V \cup E \cup C^k} h^{(i)}(v) \\right).$$
> >
> > - **DSS.** DSS is a subgraph GNN. From a graph $G$ it extracts a set of subgraphs $\\mathcal{S}$. For every subgraph $S \\in \\mathcal{S}$ and each node in that subgraph $v \\in V(S)$ it computes an embedding $h^{(i)} (v,S)$. Then,
> > $$\\text{M}^{(i)} = \\frac{1}{\\sum_{S \\in \\mathcal{S}} |V(S)|} \\sum_{S \\in \\mathcal{S}} \\sum_{v \\in S(V)} h^{(i)} (v,S).$$
> >
> > - **L2GNN.** L2GNN computes embeddings $h^{(i)}(u,v)$ for every pair $(u,v)$ of nodes $u \\in V, v\\in V$. Hence,
> > $$\\text{M} = \\frac{1}{|V|^2} \\sum_{u, v \\in V}  h^{(i)}(u,v).$$
> >
> >
> >
> > We believe that this is the simplest KD method that is agnostic to the expressive GNNs (besides label smoothing).
> >
> > > Given that you select teachers based on test set performance, how can you ensure that your reported results for GIN+KD are not optimistically biased due to information leakage from the test set?
> >
> > Our goal is not to give accurate performances of MPNNs on these datasets, but to show that we can significantly boost MPNN performance. Thus while it is true that our selection optimistically biases KD our core message still holds.

---

### Official Review · Reviewer_rJ44 · 2025-10-31

**Soundness:** 3
**Presentation:** 3
**Contribution:** 3
**Rating:** 6
**Confidence:** 3

**Summary:**

The paper distills a variety of expressive teacher GNNs into GIN MPNN students, showing 50–100\% gap closure with 2–33× inference speedups, especially on molecular benchmarks where WL-level expressivity often suffices. For classification tasks, soft-label distillation consistently lifts GIN, while for regression, layer alignment plus teacher-labeled augmentation helps on ZINC and alchemy but provides limited or no gains on some datasets (e.g., QM9). Findings suggest optimization and inductive biases, rather than inherent expressivity limits, are primary drivers of observed performance gaps in many molecular settings, making KD a practical path to near-teacher accuracy with MPNN efficiency.

**Strengths:**

1. Substantial gains with simple KD and strong runtime benefits across multiple datasets.
2. Nuanced reframing of the expressivity-performance link with empirical evidence on molecules.
3. Diverse teachers and a clear GIN baseline improve comparability and insight.

**Weaknesses:**

1. Claims appear broad relative to molecular scope; tone down “invalidates” beyond covered domains.
2. Need more details on budgets, fairness, and variance to support generality.
3. Limited guidance on when KD fails due to true expressivity gaps, especially off molecules.

**Questions:**

1. Report calibration, robustness, and OOD performance of distilled models.
2. Sensitivity to student size, augmentation, and KD loss; consider cross-validation.
3. Share checkpoints and pipelines to support reproducibility.

---

> ### Author Response · Authors · 2025-11-21
>
> Thank you for your review.
>
> > Claims appear broad relative to molecular scope; tone down “invalidates” beyond covered domains.
>
> We have adapted the paper to better match or focus on molcules. In particular, we have explicitly added our molecular focus to the title. Combined with our changes to abstract, introduction and conclusion our tone should now fit our scope.
>
> > Need more details on budgets, fairness, and variance to support generality.
>
> On budgets: for this we have written a general reply about the cost of knowledge distillation. We argue that it only adds a small computational overhead (unless extensive data augmentation is performed). What did you have in mind for fairness?
>
> On variance. Consider the distillation results in Table 3 and observe that GIN+KD is always among the models with the lowest standard deviation.  Furthermore, consider Figure 4 where we demonstrate that MOLHIV models trained with KD are more stable across training than MPNNs without KD and the teacher.
>
> > Limited guidance on when KD fails due to true expressivity gaps, especially off molecules.
>
> On molecular data, KD seems to fail most commonly on datasets with regression tasks. An intuitive explanation is that regression is a more nuanced and difficult task than (binary) classification. To the best of our knowledge we could not identify any non-molecular dataset where expressive GNNs significantly outperform MPNNs.

---

> > ### Author Response · Authors · 2025-11-21
> >
> > ## Questions
> >
> > > Report calibration, robustness, and OOD performance of distilled models.
> >
> > For OOD calibration, we measured this on MOLHIV  for a teacher GSN model and a GIN model trained with KD and without. We report the Expected Calibration Error from Guo et al.,  ([On Calibration of Modern Neural Networks](https://arxiv.org/pdf/1706.04599); Guo et al.; ICML 2017) (smaller is better):
> > - GIN (without KD): 0.0210
> > - GIN (with KD): 0.0153
> > - Teacher (GSN): 0.0151
> >
> >
> > We can see that KD increases the calibration of GIN and closely matches it to the calibration of the teacher. For robustness, what kind of measurements did you have in mind? Finally, we would like to note that especially our results from OGB datasets (molhiv, molesol, molbace, moltox21) are OOD as they use a scaffold split (see [Open Graph Benchmark: Datasets for Machine Learning on Graphs](https://arxiv.org/pdf/2005.00687); Hu et al., NeurIPS 2020):
> > > We adopt the scaffold splitting procedure that splits the molecules based on their
> > > two-dimensional structural frameworks. The scaffold splitting attempts to separate structurally different molecules into different subsets, which provides a more realistic estimate of model performance
> > > in prospective experimental setting. [...] we aim to standardize the scaffold split by adopting its most challenging version where test molecules are maximally diverse
> >
> > > Sensitivity to student size, augmentation, and KD loss; consider cross-validation.
> >
> > - **Student size.** Note that in all experiments (except on ZINC), the student has the same number of layers and embedding dimensions as the teacher. Thus, KD distillation is not dependent on the size of the student. However, for ZINC we notice that GNN performance increases with the number of MPNN layers ($0.109$ MAE with 23 layers vs $0.129$ MAE with 5 layers). This is particularly noteworthy, as oversmoothing makes it difficult to train such deep GNNs. However, with distillation it is possible.
> > - **Augmentation.** We only experiment with data augmentation for regression datasets, as label smoothing suffices to achieve strong performance on classification datasets. On regression datasets, the connection between the amount of data augmentation and student performance seems to be task and dataset dependent. On ZINC, student performance increases with augmented data and it seems that there is no point at which more data is harmful. On molesol, a data augmentation factor of ~10 (so 10x the original training set on top of the original training set) achieves the best performance and performance decreases with more data. On both QM9 and alchemy we observe no improvement with data augmentation.
> > - **KD loss.** We only have a trade-off between KD loss and prediction loss when we perform layer alignment (for regression tasks). Here are the results from us tuning KD loss balancing factor $\\alpha \\in \\{ 0.01, 0.1, 1, 10 \\}$ and the data augmentation factor $m \\in \\{ 0, 10, 50 \\}$
> > 	- On molesol and ZINC, the loss balancing factor has little impact on predictive performance. In both cases, the data augmentation factor is more important. For molesol, $m = 10$ gives the best performance. For ZINC $m = 50$ gives the best performance which motivates our experiments into further scaling data augmentation on ZINC.
> > 	- On alchemy, the loss balancing factor has a large impact on the predictive performance. Here $\\alpha = 1$ achieves the best performance. In contrast, data augmentation is less important on alchemy, interestingly the best performing student uses no data augmentation $m = 0$.
> >
> >
> > > Share checkpoints and pipelines to support reproducibility.
> >
> > After publication we will share everything that is needed to reproduce our work. Note that our code and documentation are already public which allows for easy reproduction.
> >
> >
> > On variance. Consider the distillation results in Table 3 and observe that GIN+KD is always among the models with the lowest standard deviation.  Furthermore, consider Figure 4 where we demonstrate that MOLHIV models trained with KD are more stable across training than MPNNs without KD and the teacher.
> >
> > > Limited guidance on when KD fails due to true expressivity gaps, especially off molecules.
> >
> > On molecular data, KD seems to fail most commonly on datasets with regression tasks. An intuitive explanation is that regression is a more nuanced and difficult task than (binary) classification. To the best of our knowledge we could not identify any non-molecular dataset where expressive GNNs significantly outperform MPNNs.

---

### Author Response · Authors · 2025-11-20
**General reply: On non-molecular datasets**

One point that was raised multiple times is an evaluation on non-molecular data. However, on such datasets expressive GNNs do not generally outperform MPNNs. We evidence this in Appendix A, where we perform a statistical analysis for three of the most commonly used non-molecular datasets (IMDB-B, IMDB-M, reddit-B). In this analysis, we  show that on these dataset expressive GNNs do not outperform MPNNs with statistical significance. Thus, these datasets are uninformative for assessing whether expressivity causes performance gaps.


We have extended our search to other non-molecular datasets (such as MalNetTiny). We would be thankful for any suggestions.


Regarding the other points raised by the reviewers, we are still in the process of formulating a reply and running some other experiments and will get back to you soon.

---

### Author Response · Authors · 2025-11-21
**# General Reply: Overview of proposed changes and new results**

We thank the reviewers again for their efforts. In this reply, we first give an overview of our changes and additional experiments.
Then, we give an explanation of scenarios that explains the connection between expressivity and knowledge distillation (KD).
Finally, we summarize the cost of performing KD and argue that it only adds a small overhead over training expressive GNNs.

Following reviewer suggestions we have made the following changes to our paper:
- **Molecular focus:** we have changed the **title** (Do Molecular Tasks Need Expressive GNNs? Distilling Expressive GNNs into MPNNs), abstract, introduction and conclusion to better reflect our focus on molecules.
- Include section giving precise details on layer alignment (see response to wjhX)
- Include section on the cost of KD (see below)
- Include section giving intuition for cases where KD does and does not help (see below)

We have also performed additional experiments which we will add to the _next_ version of the paper (details in the comments to the respective reviewer):

- Calibration measurement of GIN + KD vs GIN vs expressive GNN on MOLHIV (reply to rJ44): KD improves calibration and allows the student to match the teacher's calibration
- Self distillation from (average) GIN to GIN (reply to wjhX): demonstrates that better regularization is not the cause of performance improvements when using KD
- Self distillation from strong outlier GIN to gin (reply to wjhX): demonstrates that the teacher does not **need** to be expressive, it just needs to generalize well

---

> ### Author Response · Authors · 2025-11-21
> **Giving intuition: Why distillation does (not) work**
>
> Multiple reviewers (wjhX, nTZ7, 5tip) raised the point that we should develop a better understanding of why knowledge distillation (KD) does (not) work.  In what follows, we describe how KD would influence different scenarios.
>
> ## Case 1: Some Graphs in the Dataset are Indistinguishable by MPNNs
>
> ### (Case 1a) Graphs are indistinguishable and expressivity is required for the learning task
> Suppose a dataset similar to the artifical expressivity datasets mentioned by reviewer 5tip. All nodes in all graphs in this dataset have the same node-degree $\Delta$, the same number of nodes, and there exist no features. The task is to count the number of cycles in each graph. In this case, an MPNN would entirely fail: as all nodes have the same degree the MPNN would compute the same node embeddings in each graph. This is evident from simply looking at the update functions of MPNNs:
> $$h_v^{(k+1)} = f_{\\text{update}} ( h_v^{(k)}, \\{\\!\\{ h_{u}^{(k)} \\mid u \\in \\mathcal{N}(v)\\}\\!\\}).$$
> Since there exist no node and edge features, we would initialize all node features to the same vector $h^{(0)}$. Hence, after one update step:
> $$h_v^{(1)} = f_{\\text{update}} ( h^{(0)}, \\{\\!\\{ h^{(0)} \\mid u \\in \\{1,\\ldots, \\Delta \\} \\! \\}\\}).$$
> Observe that after one update step the embedding of every $v$ is independent from the graph structure (by induction this also holds for all subsequent layers). It follows that in this case, MPNNs are unable to solve the task at hand. Notably, even with KD, MPNNs are still unable to solve this task: KD does not stop the MPNN from predicting the same embedding for every node.
> KD is merely a different learning task and does not influence the architectural restrictions of MPNNs leading to identical node representations for all nodes.
>
> ### (Case 1b) Graphs are indistinguishable but expressivity is not required for the learning task
> Suppose an arbitrary graph dataset where the task is to predict a function that only depends on tree-homomorphisms. Since WL and MPNNs can count tree homomorphisms they can represent this function even if (some or even all) graphs are indistinguishable, as any two MPNN-indistinguishable graphs must have the same function value. KD may improve empirical performance of MPNNs by providing additional labeled data and/or richer signals to learn from.
>
> ## Case 2: All Graphs in the Dataset are Indistinguishable by MPNNs
>
> ### (Case 2a) Graphs are distinguishable but expressivity is required for the learning task
> Consider the following dataset. Every graph contains two disjoint subgraphs of:
> - (1) The first sugraph is either two 3-cycles or a 6 cycle.
> -  (2) The second subgraph is a tree that is unique in this dataset.
>
> The task is to predict whether the combined graph contains a 3-cycle or a 6-cycle. Observe that because of (2), all graphs in this dataset are pairwise distinguishable by MPNNs or 1-WL. However, still the MPNN is completely unable to solve this task as it cannot distinguish the two graphs in (1) --- this is one of the famous 1-WL counterexamples. Hence, MPNNs can fit the training dataset (even perfectly) but fail to generalize to unseen data. Even when performing KD, MPNNs will still be entirely unable to distinguish the two cases in (1). Hence, even though the graphs are distinguishable MPNNs with KD cannot generalize.
>
> ### (Case 2b) Graphs are distinguishable and expressivity is not required for the learning task
>
> Real-world graphs very rarely are as "sanitized" as the graphs in Case 2a: while molecules contain cycles there are often tree-like structures attached to nodes in the cycles.
> Let us now consider the same dataset as in Case 2a, but instead of having a disjoint tree as a subgraph, we will have multiple unique trees that we attach to the vertices in the cycles.
> While MPNNs in general are unable to count cycles (see [Can Graph Neural Networks Count Substructures?](https://proceedings.neurips.cc/paper/2020/file/75877cb75154206c4e65e76b88a12712-Paper.pdf); Chen at al.; NeurIPS 2020) in this case the MPNN can learn a function that can detect the cycles by exploiting the fact the the trees make the vertices in the cycle detectable.
> However, as this task is very hard, the MPNN might never learn this function. KD can make learning this function easier!

---

> > ### Author Response · Authors · 2025-11-21
> >
> > ### What does this mean
> > On the one hand, we see that KD **cannot** yield an improvement in generalization performance of MPNNs in the examples where beyond-MPNN-expressivity is required for the learning task (Cases 1a and 2a).
> > On the other hand, KD **can** yield an improvement in generalization performance of MPNNs if the learning task at hand is in fact simple enough to be (in principle) learned by MPNNs, as in Case 1b and 2b.
> > That is, in cases where the target function is representable by MPNNs (Case 1b and 2b), KD may improve generalization, while KD is unlikely to increase generalization performance when this is not the case (Case 1a and 2a). Here the MPNN is forced to give identical embeddings to the key substructures. Distillation cannot break this architectural constraint, so the student cannot represent the teacher’s function.
> >
> > This behavior is independent of the question whether the graphs in the dataset can be distinguished by MPNNs, or not. Given that most molecular graphs in the common benchmark datasets can be distinguished by 1-WL (see [1-WL expressiveness is (almost) all you need](https://arxiv.org/pdf/2202.10156); Markus Zopf; IJCNN 2022) the experiments in our paper (approximately) correspond to case 2.
> > Similar to our Case 2b, real-world molecules have trees attached to cycles that make tasks related to cycles easier.
> > Furthermore, they have node and edges features that can provide additional informations about cycles.
> > Given that many molecular properties depend on cycles with attached trees or informative features, KD helps MPNNs learn functions that exploit these structures.

---

> ### Author Response · Authors · 2025-11-21
> **The cost of performing knowledge distillation**
>
> Two reviewers (rJ44, 5tup) have asked questions connected with the cost of knowledge distillation and its fairness. We give an extensive overview of the runtime costs of training our models with knowledge distillation and expensive teachers.  Overall, knowledge distillation is efficient and only adds a small overhead on top of tuning and training the expressive teachers. The following numbers are per dataset.
> - **Hyperparameters Teacher:** The hyperparameters of every teacher are tuned. This means we train 48 teacher models
> - **Hyperparameters Student:** For both regression and classification we do not tune any of the MPNN hyperparameters (such as number of layers), instead we copy the hyperparameters of the best teacher. On regression tasks, we tune the loss balancing factor and the data augmentation factor. In total, we train 9 student models for regression tasks (not counting evaluation and further experimentation). For classification, we only train a single student model as we tune no distillation hyperparameters. Thus, knowledge distillation does not require significantly more hyperparameter tuning.
> - **Data labeling:**  The cost of data labeling is negligible, as we only need to run a single forward pass of the expressive GNNs and can even disable gradient computation. Labeling all 10'000 graphs in the training set of `ZINC` with L2GNN takes 30 seconds on a single consumer-grade GPU (RTX 3080). We would like to point out that our labeling code is inefficient as it does not make use of batching multiple graphs and thus, labeling could easily be speed up by a factor of 10 - 100.
> - **Data augmentation:** Similarly to data labeling this is negligible (3 seconds for 10'000 graphs based on ZINC on a single CPU core). Furthermore, it can be parallelized over multiple CPU cores.
> - **Training under teacher supervision:** Since embeddings and logits are pre-computed, the cost of supervision is also negligible. In the case of classification without layer alignment, distillation is as fast as training without distillation. For regression with layer alignment, the only difference is the adapted loss function which means that in practice the difference to training without distillation is small (less than 10% difference). We measured this for 100 epochs of  training and evaluation on ZINC over 5 runs:
> 	- With knowledge distillation (layer alignment, no data augmentation): $97.1 \pm 0.5$ seconds
> 	- Without knowledge distillation: $90 \pm 0.6$ seconds
>
> In summary, we can see that distillation adds almost no overhead when compared to training expressive GNNs. We would like to point out that there is one exception to the efficiency of distillation: the case of training with significantly more data than the teacher (such as 200x ZINC). In this case, the training does take significantly longer. However, this is not because distillation is slow, but because of the large amount of data. We have included this information in Appendix D of our paper.

---

> ### Author Response · Authors · 2025-12-03
> **Supporting KD Intuition with Experimental Evidence**
>
> We present a series of tasks that demonstrate our intuitive argument empirically. In all cases, we use GIN as a student and GRIT as a teacher. All models have the same hyperparameters: 5 layers, 16 embedding dimensions, 0.4 drop out rate. All our datasets contain 3000 graphs split 20%/40%/40% into training, validation and test sets. Our setting is based on our observation that on our synthetic datasets, models tend to overfit and results can be noisy. Thus, we train small models with a large drop out rate for regularization. Furthermore, we use large validation and test sets to estimate the models generalization performance.
>  Our setup is identical for every case (we only swap the dataset and task). All our tasks are classification tasks.
>
> * We train the baseline GIN 20 times
> * We train the teacher a single time.
> * We perform KD from the teacher to GIN 20 times using only label-smoothing
> * We report the averaged results
>
>
> ### (Case 1) Pairwise indistinguishable
> We simplify Case 1 to the situation where all graphs in a dataset are indistinguishable by 1-WL and investigate whether we can solve tasks that require expressivity. This case is similar to the 1-WL hard datasets requested by reviewer 5tup. Indeed, the dataset CSL requested by 5tup is 2-regular and BREC [1] contains a category with regular graphs.
>
> The task is to predict whether a 3-regular graph contains at least 10 triangles. We generate 3000 random 3-regular graphs with 20 nodes each. Since all nodes have identical degree and features, MPNNs compute identical embeddings for all graphs, making triangle counting impossible. As our Case study predicted, KD provides no benefit: baseline GIN achieves ~52% accuracy (random guessing), and KD-GIN shows no improvement, confirming that distillation cannot overcome fundamental expressivity limitations. The results are (accuracy, larger is better):
>
> * MPNN: $52 \pm 0.01$
> * Expressive GNN: $100$
> * MPNN + KD: $52 \pm 0.01$
>
>
> ### (Case 2) Pairwise distinguishable
> Next, we move to graphs that are pairwise distinguishable. While our initial Cases 2a and 2b where artificially created examples we will present a more unified setting here. We use a  dataset of 3000 Erdős-Rényi random graphs with n=50 nodes and edge probability p=0.138. The task is binary classification: does the graph contain at least 50 triangles? Due to different degree sequences these graphs are pairwise distinguishable. However, triangle counting is a task that the MPNN is unable to do  [2]. First (Case 2a), we show that in this case KD provides no measurable improvement. Afterwards (Case 2b), we make this task easier for the MPNN by introducing node and edge features that slightly correlate with triangle counts meaning that in principle the MPNN should be able to solve this task. However, our experiments demonstrate that the vanilla MPNN is unable to pick up on this correlation which means that performance stays approximately constant. However, KD allows the MPNN to detect this correlation and boost performance.
>
> #### (Case 2a) Pairwise Distinguishable + Expressivity Required for Task
> Results show KD provides no benefit when the task fundamentally requires expressivity the student architecture lacks. The results are (accuracy, larger is better):
>
> * MPNN: $84.5 \\pm 0.4$
> * Expressive GNN: $94.0$
> * MPNN + KD: $84.4 \\pm 0.6$
>
>
> #### (Case 2b) Pairwise Distinguishable + Features that help with the task
> We add node and edge features that correlate with the triangle counts but have added noise on top.
> Details. Each node feature is a combination of local triangle participation, normalized degree, and a small random global signal (a single 0/1 “global bit” sampled once per graph that shifts all node features slightly in the same direction). Edge features count the number of triangles the edge participates in, weighted slightly toward a constant baseline and perturbed with small Gaussian noise. All features are quantized into 10 integer bins to allow the use of dictionary encoders.
> Results. We can see that the performance of the baseline MPNN slightly improves, which is expected. The additional noisy information leads to decrease in performance of the expressive GNN. Finally, KD allows the MPNN to make use of the additional information and significantly boosts the performance of the MPNN (p = 0.05 using a Welch t test)!
>
> * MPNN: $85 \\pm 1.1$
> * Expressive GNN: $88.5 $
> * MPNN + KD: $\mathbf{87.1} \\pm 0.4$
>
>
> ### Implications
> We can see that knowledge distillation is helpful when there is enough information for the MPNN to detect. Notable, in molecules there is correlation between node and atom types and the graph structure. This mirrors our case with distinguishable graphs and helpful features in which can demonstrate the KD boosts the predictive performance.
>
> [1] Wang & Zhang, An Empirical Study of Realized GNN Expressiveness, ICML 2024
> [2] Chen et al., Can Graph Neural Networks Count Substructures?, NeurIPS 2020

---

### Author Response · Authors · 2025-12-03
**Rebuttal Summary**

We provide a summary of the reviewers' main concerns and how we have addressed them.


* **Claims beyond molecules:** All reviewers have pointed out that our claims read like they extend beyond the domain of molecules. We have rewritten the abstract, introduction and conclusion to better represent our molecular focus. Furthermore, we have changed our title to: _Do Molecular Tasks Need Expressive GNNs? Distilling Expressive GNNs into MPNNs_.
* **Developing a better understanding of when KD does (not work):** All reviewers have asked for more exploration of why KD works:
    * We have performed experiments to address two questions raised by reviewer wjhX. First, does KD improve the students performance by working as a regularizer? We observe that self-distillation from an MPNN with average performance does not improve predictive performance. This provides evidence that the performance improvements caused by KD are not primarily due to regularization.
Second, does KD require an expressive teacher? We observe that we can perform KD from an outlier MPNN with strong predictive performance into another MPNN to boost predictive performance. Thus, KD requires no expressive teacher for strong performance improvements.
Furthermore, at the request of reviewer rJ44 we have measured the OOD calibration of models trained via KD. We find that KD improves OOD calibration (baseline MPNN: 0.0210 vs. MPNN + KD: 0.0153 Expected Calibration Error [1]). In particular, students and teachers have almost identical calibration (MPNN + KD: 0.0153 vs Teacher: 0.0151 Expected Calibration Error).
    * We provided intuition for when KD does (not) work across four cases covering indistinguishable/distinguishable graphs and tasks requiring/not requiring expressivity (see General Reply: Supporting KD Intuition with Experimental Evidence).
    * We conducted experiments on synthetic datasets to empirically support this intuition. They show that KD provides no benefit if the tasks strictly require > $1$-WL expressivity. However, if tasks are MPNN-representable but involve only subtle feature-structure correlations, KD enables significant learning improvements. In our view this explains performance gains on molecular benchmarks: In molecular data, node/edge features correlate with structural properties, but MPNNs struggle to learn these correlations without KD's training signal.
* **Understanding the costs of knowledge distillation:** Two reviewers (rJ44, 5tup) were interested in more details about the (runtime) cost of KD and the overhead it introduces. We have analyzed all aspects of KD (see The cost of performing knowledge distillation) and added this to the appendix. In particular:
    * Students trained via KD require less tuning of model specific hyperparameters.
    * The cost of data augmentation and labeling is neglible (~30 seconds for 10'000 graphs).
    * Training under KD supervision adds little training time (less than 10% speed decrease).
* **Clarity of the "Layer Alignment" Method:** Reviewer wjhX stated that our definition of the layer alignment method is vague. We have formalized its definition and added it to the paper.


[1] Guo et al., _On Calibration of Modern Neural Networks_; ICML 2017

---

### Meta-Review · Area_Chair_RWHD · 2026-01-04

**Summary:**

The reviewers' concerns primarily focus on the paper's theoretical framing and the generalizability of its claims regarding Graph Neural Network (GNN) expressivity. While the reviewers acknowledged the practical utility of the proposed knowledge distillation (KD) framework for molecular tasks , there was significant resistance to the authors' central thesis that the performance gap between expressive GNNs and Message Passing Neural Networks (MPNNs) is primarily due to optimization rather than expressivity limitations. Reviewer 5tup, in particular, argued that molecular benchmarks are often "tree-like" and 1-WL distinguishable, meaning MPNNs already possess sufficient theoretical expressivity for these specific tasks. Consequently, the paper demonstrates that KD improves optimization in regimes where expressivity is not a bottleneck, rather than proving expressivity is irrelevant generally. Additionally, reviewers raised concerns about the lack of formal guarantees that the student model inherits the teacher's power and the limited evaluation on non-molecular datasets.

**Reviewer Concerns:**

Reviewer Concerns Addressed by Rebuttal:


1. Scope and Title: The authors successfully addressed concerns regarding the over-broadness of their claims by narrowing the scope to molecular tasks and changing the title to "Do Molecular Tasks Need Expressive GNNs?".



2. Operational Costs: Questions regarding the computational cost and overhead of the distillation process were addressed with a detailed analysis showing negligible overhead for labeling and data augmentation.



3. Methodological Clarity: The vague definition of "layer alignment" was formalized and added to the appendix.



4. Why KD Works: The authors provided additional experiments investigating KD as a regularizer and the necessity of expressive teachers.

Outstanding:

1. Theoretical Generalizability: Reviewer 5tup's concern that the results rely on the specific structural properties of molecules (tree-like, 1-WL distinguishable) remains a critical outstanding issue. The authors' counter-argument that real-world data falls into specific "cases" did not convince the reviewer to change their score, as the student MPNN remains 1-WL bounded at inference time regardless of the training signal.


2. Evaluation on Non-Molecular Datasets: Reviewer nTZ7 and wjhX requested evaluation on non-molecular benchmarks to test generalizability. The authors argued they could not find non-molecular datasets where expressive GNNs significantly outperform MPNNs, which leaves the scope of the contribution strictly limited to the molecular domain.

**Reviewer Scores:**

Reviewer rJ44 (6): Likely would remain a 6. They found the findings sound and the rebuttal regarding calibration and costs sufficient, having noted the strengths in runtime benefits.

Reviewer wjhX (6): Likely would remain a 6. While they appreciated the practical utility , they strongly cautioned against the "death of expressivity" narrative. The authors' pivot to a molecular-specific title mitigates this, but the reviewer might still view the theoretical contributions as overstated.


Reviewer nTZ7 (4): Likely would remain a 4 or move to a 5. This reviewer requested experiments to support the authors' intuitions. Although the authors provided synthetic experiments in a late reply, the reviewer did not confirm these were sufficient to justify an improved score.


Reviewer 5tup (4): Would remain a 4. This reviewer explicitly stated after the rebuttal that they would maintain their score , indicating that the rebuttal failed to address their fundamental concern about the confounding factor of molecular graph structure (1-WL sufficiency).

---

### Decision · Program_Chairs · 2026-01-26

Reject